# Feasibility of a pulmonary rehabilitation programme for patients with symptomatic chronic obstructive pulmonary disease in Georgia: a single-site, randomised controlled trial from the Breathe Well Group

Mariam Maglakelidze,[1,2] Ia Kurua,[1] Nino Maglakelidze,[1] Tamaz Maglakelidze,[1,3] Ivane Chkhaidze,[1,4] Ketevan Gogvadze,[1] Natia Chkhaidze,[5] Helen Beadle,[6] Kelly Redden-Rowley,[7] Peymane Adab,[8] Rachel Adams [8] Chunhua Chi,[9] KK Cheng,[8] Brendan Cooper,[10] Jaime Correia-de-Sousa,[11,12] Andrew P Dickens [8,13] Alexandra Enocson,[8] Amanda Farley,[8] Nicola K Gale,[14] Sue Jowett,[8] Sonia Martins,[15] Kiran Rai,[8] Alice J Sitch,[8,16] Katarina Stavrikj,[17] Rafael Stelmach,[18] Alice M Turner,[8] Sian Williams,[12] Rachel E Jordan,[8] Kate Jolly[8]

For numbered affiliations see end of article.

**Correspondence to**
Tamaz Maglakelidze;
tmaglak@gmail.com

## ABSTRACT

**Objectives** To assess the feasibility of delivering a culturally tailored pulmonary rehabilitation (PR) programme and conducting a definitive randomised controlled trial (RCT).

**Design** A two-arm, randomised feasibility trial with a mixed-methods process evaluation.

**Setting** Secondary care setting in Georgia, Europe.

**Participants** People with symptomatic spirometry-confirmed chronic obstructive pulmonary disease recruited from primary and secondary care.

**Interventions** Participants were randomised in a 1:1 ratio to a control group or intervention comprising 16 twice-weekly group PR sessions tailored to the Georgian setting.

**Primary and secondary outcome measures** Feasibility of the intervention *and* RCT were assessed according to: study recruitment, consent and follow-up, intervention fidelity, adherence and acceptability, using questionnaires and measurements at baseline, programme end and 6 months, and through qualitative interviews.

**Results** The study recruited 60 participants (as planned): 54 (90%) were male, 10 (17%) had a forced expiratory volume in 1 second of ≤50% predicted. The mean MRC Dyspnoea Score was 3.3 (SD 0.5), and mean St George's Respiratory Questionnaire (SGRQ) 50.9 (SD 17.6). The rehabilitation specialists delivered the PR with fidelity. Thirteen (43.0%) participants attended at least 75% of the 16 planned sessions. Participants and rehabilitation specialists in the qualitative interviews reported that the programme was acceptable, but dropout rates were high in participants who lived outside Tbilisi and had to travel large distances. Outcome data were collected on 63.3% participants at 8 weeks and 88.0% participants at 6 months. Mean change in SGRQ total was −24.9 (95% CI −40.3 to −9.6) at programme end and −4.4 (95% CI −12.3 to 3.4) at 6 months follow-up for the intervention group and −0.5 (95% CI −8.1 to 7.0) and −8.1 (95% CI −16.5 to 0.3) for the usual care group at programme end and 6 months, respectively.

**Conclusions** It was feasible to deliver the tailored PR intervention. Approaches to improve uptake and adherence warrant further research.

**Trial registration number** ISRCTN16184185.

## STRENGTHS AND LIMITATIONS OF THIS STUDY

⇒ This is the first published pulmonary rehabilitation (PR) trial undertaken in Georgia.

⇒ The intervention was culturally tailored for a middle-income country, having been selected through a structured prioritisation process involving policy makers, clinicians and patients.

⇒ The 63% follow-up in the intervention group at 8 weeks affects interpretation.

⇒ A post hoc per-protocol analysis explored the outcomes in intervention participants who attended at least 50% of the PR sessions, showing a consistent pattern of improvements in health-related quality of life.

⇒ Recruitment through primary care proved challenging due to patients with a diagnosis of chronic obstructive pulmonary disease from primary care not fulfilling diagnostic criteria on spirometry.

## INTRODUCTION

Chronic obstructive pulmonary disease (COPD) is a progressive chronic inflammatory lung disease that has a significant health and economic burden, with high healthcare

costs arising largely from hospital admissions for exacerbations.[1–3] It is currently the third leading cause of death worldwide[2] with 251 million cases reported globally in 2016.[4] Although most of these deaths occur in low-income and middle-income countries (LMICs),[5] most research on COPD management has been undertaken in high-income countries. The burden of chronic respiratory disease (CRD) is a considerable challenge for Georgia's healthcare system with the mortality attributable to CRD about 4% in Georgia,[6] and as elsewhere in the world, rates are increasing mainly due to air pollution (outdoor and indoor from solid fuel cooking and heating);[6 7] and high rates of tobacco use[2] (about a third of adults are current smokers). The availability of pharmacotherapy is limited due to resource constraints, making pharmacological management difficult.[8]

In addition to the direct health and healthcare burden, COPD negatively impacts the health-related quality of life of people living with the condition.[9] There is growing evidence that pulmonary rehabilitation (PR) is an effective and cost-effective therapeutic intervention to improve COPD symptoms, patients' quality of life[10] and to reduce risk of death when delivered early following hospital admission for exacerbation.[11] However, evidence of the effectiveness of PR for COPD in patients from LMICs is extremely limited; barriers to provision include infrastructure for diagnosis, essential drug availability, lack of skilled health professionals and overall healthcare priorities, which limit management options[12] and few studies have adapted PR to the local context in LMICs.[12–14] Additionally, local adaptation (or tailoring) has been reported as an enabler of PR in LMICs.[15] There are currently no PR services offered to patients in Georgia, a middle-income country. In this context, where resources are limited, yet COPD is a major burden, research is needed to evaluate whether offering PR is feasible and to assess whether the intervention would have similar effects to those observed in other settings. This study evaluated the feasibility of a future trial to assess the effectiveness of a culturally tailored PR programme in Georgia.

## METHODS
### Study design and participants
The study was a two-arm, non-blinded, randomised feasibility trial of an adapted PR programme, with a mixed-methods process evaluation conducted in the Chapidze Emergency Cardiology Center, Tbilisi. Participants with an Medical Research Council (MRC) Dyspnoea Score ≥2, and a spirometry-confirmed COPD diagnosis, defined as a postbronchodilator forced expiratory volume in 1 second ($FEV_1$)/Forced Vital Capacity (FVC) ratio <0.70, were deemed eligible for the study. Exclusion criteria were the presence of musculoskeletal or neurological conditions preventing exercise, unstable cardiovascular disease (eg, unstable angina, aortic valve disease, unstable pulmonary hypertension), severe cognitive impairment, severe psychotic disturbance, active contagious infectious disease and very poor balance.

### Sample size
We aimed to recruit 60 participants. The sample size was chosen to enable estimation of feasibility outcomes with reasonable precision.[16] A follow-up rate of 80% could be estimated with a precision of 68%–89% (binomial exact 95% CI).

### Recruitment and randomisation
Participants were recruited (December 2018 to May 2019) from six primary care facilities and three secondary care facilities in Tbilisi; all facilities provided services to patients without geographical restriction. An information brochure about the programme was distributed among doctors in primary care facilities and doctors from the Georgian Respiratory Association working in secondary healthcare. These facilities provided contact details of patients with COPD, who were contacted via telephone. The MRC Dyspnoea Score[17] was completed over the telephone and those with a score of ≥2 were invited to the spirometry eligibility assessment. Before spirometry, patients were given a study participant information leaflet. Postbronchodilator spirometry was performed according to American Thoracic Society (ATS) and European Respiratory Society (ERS) 2005 guidelines[18] using a Spirolab III spirometer and over-read by a member of the research team (AE or BC). Eligible participants were asked to give informed, written consent to participate in all aspects of the trial, including qualitative interviews. Randomisation to one of the two groups in a 1:1 ratio was performed using an electronic database (REDcap).[19 20]

### Intervention
The intervention was adapted from PR programmes delivered in the UK, and following UK and ERS/ATS guidance.[21 22] Three focus groups took place in Tbilisi, Georgia and involved people with COPD (n=6), family practitioners and respiratory physicians (n=6), and rehabilitation specialists (n=7) who discussed potential components for the PR programme. Their views were considered alongside the APEASE criteria[23] (Affordability, Practicality, Effectiveness and cost-effectiveness, Acceptability, Side-effects/safety, Equity). Adaptations were made to the programme, taking into consideration contextual, cultural and resource needs in Georgia. Given the high literacy rates of the population, an information booklet about COPD, covering the 16 topics included in the PR education sessions, was also produced in the local language. The main adaptations included recognition of the importance of including family members, by inviting them to attend a prerehabilitation information session and the educational talks. The educational sessions did not include discussions of end of life or the potentially terminal nature of COPD as requested by focus group participants.

A prerehabilitation one-to-one assessment was booked at a time convenient for the patient. The aim of this assessment was to orientate the patient to the PR intervention and tailor their exercise programme. During this visit, patients were given a PR information booklet, created for the trial by the research team. The distance walked in the incremental shuttle walk test (ISWT),[24] Borg Rating of Perceived Exertion and Modified Borg Breathlessness Scales[25] were used to prescribe the intensity of the exercise component of the PR programme. Participants had pulse oximetry measured during the ISWT and were prescribed 65%–85% of their maximal baseline ISWT performance (lower in deconditioned patients). Changes were not made to patients' medication.

The adapted PR programme drew on international guidance[21 22] and consisted of two main components: 1.5 hours of exercise which took place in a hospital rehabilitation gym and 15–20 min of education on the management of COPD covering 16 topics (see online supplemental table S1 for additional information on intervention content). The description, content and frequency of these components were determined during the intervention development phase of this project (cultural tailoring). The programme took place twice weekly for an 8-week period as a rolling programme (ie, new patients were introduced at any point).

The intervention was delivered by two rehabilitation specialists (physiotherapists), who had not previously delivered PR. They were trained by UK respiratory physiotherapists to deliver PR over two visits and received a total of 4 days of training. The training topics are detailed in online supplemental table S2.

### Usual care group
Participants randomised to the control group received usual care from their primary care doctor and/or pulmonologist. They were offered a delayed 1.5–2 hours educational session delivered by pulmonologists and specialists once the final (6 months) follow-up was complete. During this session, participants received the same PR educational booklet as the intervention group.

### Outcome measures
Data were collected from patients by the research team members at the end of the PR programme (8 weeks) and at 6 months, at a clinical assessment; participants who were unable or unwilling to attend the assessment were invited to complete the questionnaires over the telephone. Follow-up was not blinded. A baseline questionnaire captured sociodemographic and health-related characteristics of the participants.

### Feasibility outcomes
The main feasibility outcomes were delivery of the PR with fidelity, acceptability to participants, recruitment rate, follow-up rate at 6 months, adherence to the intervention, ability to carry out trial procedures, feasibility of methods to measure costs of PR, other COPD-related healthcare utilisation (including number of hospital admissions since enrolment) and patient-incurred costs.

### Secondary outcomes
Secondary outcomes included the primary outcome of a future definitive randomised controlled trial (RCT), which was the St George's Respiratory Questionnaire (SGRQ)[26] at 6 months (total, symptoms, activity and impact). Other secondary outcome measures at 8 weeks and 6 months were: SGRQ at programme end, exercise capacity measured by the ISWT (metres),[24] smoking status validated by cotinine, COPD Assessment Test (CAT),[27] self-reported physical activity (International Physical Activity Questionnaire - IPAQ),[28] self-efficacy measured by the Stanford Self-Efficacy Scale,[29] depression (Patient Health Questionnaire-9 (PHQ-9))[30] anxiety (generalised anxiety disorder-7 (GAD-7))[31] and self-reported number of exacerbations (in the last 6 months).

To assess the feasibility of collecting outcome data for a future health economic analysis we collected the EQ-5D-5L,[32] self-reported COPD-related healthcare utilisation and participant-incurred costs from an open question in the 8-week questionnaire. In order to determine whether it would be possible to estimate the cost of PR in a future trial we recorded the number of sessions attended by each participant and total number of sessions delivered.

### Process evaluation
Process measures to determine the feasibility of intervention delivery and study processes included number of PR sessions attended, fidelity of delivery and participant engagement assessed by observation of 16 sessions. Completion of PR was defined as attendance at 75% of the sessions.[33]

One rehabilitation session was observed by a member of the research team and exercise sheets were analysed to make sure they were delivered with fidelity. Fidelity was defined as evidence of increasing exercise prescriptions and recorded activities over the PR course for three aerobic and three strength exercises.

We telephoned patients who had agreed to be contacted for interview, using purposive sampling to increase the mix of sociodemographic characteristics, severity of COPD and differing levels of engagement with the PR programme. After completion of the programme, semistructured interviews were conducted with nine participants (who attended between 2 and 16 PR sessions) and the two rehabilitation specialists to assess their experience of it. Participant interviews were conducted by telephone, and face to face with the rehabilitation specialists. Structured topic guides (see online supplemental table S3) were used and refined iteratively as themes emerged. The average interview duration was 35 min for the patient participants and 33 min for the rehabilitation specialists. Sample size was determined by thematic saturation.[34]

## Data analysis

Analyses concentrated on available data only, with no attempt made to impute missing values. Analyses commenced once the last participant completed follow-up. Participant demographics were tabulated to understand the population recruited and whether recruitment reflected the sociodemographic profile of people with COPD in Georgia. We report recruitment and follow-up rates, with 95% CIs, overall and by study arm. Although the trial was not powered to detect a difference between intervention and wait-list control, we calculated the mean change in SGRQ at PR end and 6 months for those allocated to each group; 95% CIs were provided for estimates obtained. We provided estimates of the secondary outcomes by group and display binary outcomes using counts and percentages, and continuous data are presented using means and SD. Since the study was not powered to detect treatment effects on clinical outcomes, p values and 95% CIs were not reported. Data were analysed using STATA/IC V.15.1.

To explore whether the lack of effect at 6 months in the intervention group was due to lack of sustained effect or lack of engagement with PR by many of the intervention participants a post hoc per-protocol analysis was undertaken. Outcomes of intervention group participants who attended at least 50% of sessions[35] (n=13) were calculated.

Audio recordings of qualitative interviews were transcribed intelligently verbatim, anonymised and analysed using content analysis.[36]

## Patient and public involvement

A research prioritisation exercise was initially conducted with patients, clinicians and policy makers; the need to find effective, affordable COPD management that suited Georgian priorities and processes was selected. A patient advisory group in the UK advised on PR design, highlighting the importance of sessions being local with minimal travel costs. The intervention adaptation had input from six patients who had COPD. The study was overseen by a Trial Steering Committee, which included a person with COPD, who was involved throughout all stages of the study, contributing to decisions by commenting on intervention process, materials, advising on recruitment approaches and dissemination of findings.

## RESULTS
### Feasibility of conducting a definitive RCT
#### Participant recruitment and follow-up

We contacted 312 patients in order to recruit 60 for the study. After meeting primary criteria for study inclusion, 159 patients underwent spirometry, of whom 74 had COPD confirmed. Sixty-three patients were assessed for eligibility and attended the baseline visit, 60 were recruited and randomised (30 intervention and 30 usual care) (figure 1). One participant withdrew from the intervention group; follow-up at 6 months was 88% across both groups. Recruitment from primary care was challenging

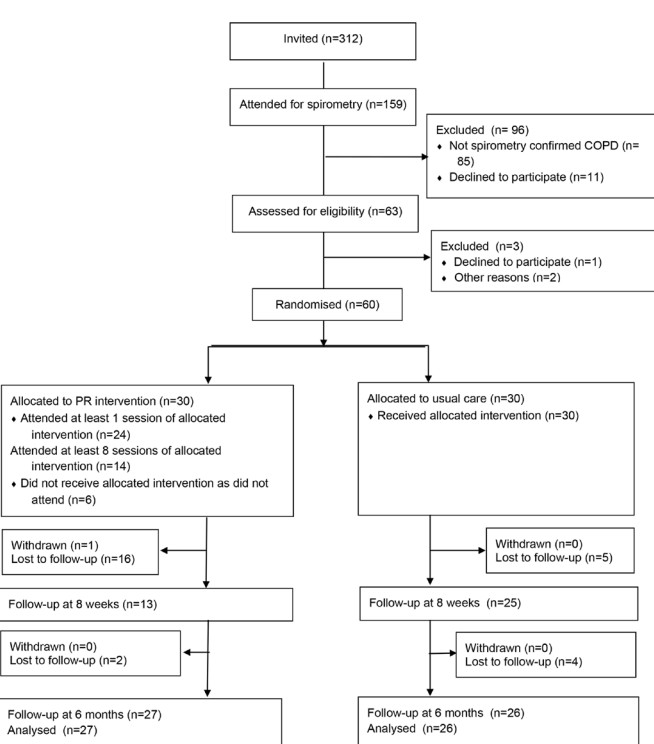

**Figure 1** CONsolidated Standards of Reporting Trials (CONSORT) flow diagram. COPD, chronic obstructive pulmonary disease; PR, pulmonary rehabilitation.

with the majority (n=53, 88.3%) of participants coming from a hospital setting. Participants who were identified in primary care and assessed for eligibility were frequently excluded following spirometry.

### Baseline characteristics of participants

Participant characteristics are detailed in table 1. Overall, 90% of participants were male, the mean age was 65 years (SD 7.95); most were Georgian (n=56; 93.3%). Twenty-three (38.3%) were current smokers and all had at least one comorbidity. Only 20 (33.3%) were in paid work. Few participants (n=10, 16.7%) were not using any COPD medication. The mean (SD) MRC Dyspnoea Score was 3.28 (0.52) and 10 (16.7%) had an FEV1% predicted of ≤50% indicating severe COPD. The mean SGRQ was 50.87 (SD 17.59).

### Follow-up rates

Follow-up rates at 8 weeks were 63.3% (38), but were much lower in the intervention arm (13; 43.3%) compared with the usual care group (25; 83.3%). Participants who did not complete follow-up at 8 weeks had similar characteristics to the full sample and did not differ by smoking status, SGRQ or PHQ-9 Score. Follow-up at 6 months was 88.3% (n=53); seven participants were administered the questionnaire by telephone, so did not undertake the ISWT, hand grip strength or cotinine measurement. One participant completed the questionnaire by telephone but attended the assessments.

**Table 1** Baseline characteristics of the patients in the intervention and usual care groups

| Characteristic | Intervention n=30 | Usual care n=30 | All n=60 |
|---|---|---|---|
| Sex, n (%), male | 26 (86.7%) | 28 (93.3%) | 54 (90.0%) |
| Age in years, mean (SD) | 64 (7.3) | 65 (8.7) | 65 (8) |
| Ethnicity, n (%) | | | |
| Georgian | 28 (93.3) | 28 (93.3) | 56 (93.3%) |
| Other ethnic group | 2 (6.7) | 2 (6.7) | 4 (6.7%) |
| Education, n (%) | | | |
| No formal qualification | 0 (0.0) | 2 (6.67) | 2 (3.3) |
| High school education | 7 (23.3) | 6 (20.0) | 13 (21.7) |
| Degree level or higher | 15 (50.0) | 14 (46.7) | 29 (48.3) |
| Other | 8 (26.7) | 8 (26.7) | 16 (26.7) |
| Living alone, n(%) | 3 (10.0) | 4 (13.3) | 7 (11.7) |
| Relationship status. n (%) | | | |
| Married and living with partner | 22 (73.3) | 22 (73.3) | 4 (73.3) |
| Never married | 3 (10.0) | 4 (13.3) | 7 (11.67) |
| Separated, but still legally married | 1 (3.3) | 1 (3.3) | 2 (3.3) |
| Divorced | 4 (13.3) | 1 (3.3) | 5 (8.3) |
| Widowed | 0 (0.0) | 2 (6.67) | 2 (3.3) |
| Employment status, n (%) | | | |
| In paid work (full-time or part-time including self-employed) | 7 (23.3) | 13 (43.3) | 20 (33.3) |
| Unemployed/looking for work | 10 (33.3) | 2 (6.7) | 12 (20.0) |
| Retired from paid work | 11 (36.7) | 11 (36.7) | 22 (36.7) |
| Looking after the family or home | 10 (33.3) | 6 (20.0) | 16 (26.7) |
| Unable to work because of my chest/lung problem | 5 (16.7) | 1 (3.3) | 6 (10.0) |
| Unable to work because of other long-term health problem | 2 (6.7) | 1 (3.3) | 3 (5.0) |
| Other | 6 (20.0) | 5 (16.7) | 11 (18.3) |
| Comorbidities—ever diagnosed, n(%) | | | |
| Diabetes | 5 (16.7) | 2 (6.7) | 7 (11.7) |
| High blood pressure | 11 (36.7) | 11 (36.7) | 22 (36.7) |
| Coronary heart disease/angina/heart attack | 6 (20.0) | 4 (13.3) | 10 (16.7) |
| Heart failure | 7 (23.3) | 7 (23.3) | 14 (23.3) |
| Stroke/mini-stroke | 0 (0.0) | 4 (13.3) | 4 (6.7) |
| Asthma | 7 (23.3) | 5 (16.7) | 12 (20.0) |
| Tuberculosis | 4 (13.3) | 2 (6.7) | 6 (10.0) |
| Osteoarthritis | 5 (16.7) | 4 (13.3) | 9 (15.0) |
| Osteoporosis | 1 (3.3) | 1 (3.3) | 2 (3.3) |
| Depression | 2 (6.7) | 2 (6.7) | 4 (6.7) |
| Other condition | 5 (16.7) | 5 (16.7) | 10 (16.7) |
| Medication use | | | |
| SABA/SAMA | 9 (30.0) | 6 (20.0) | 15 (25.0) |
| LABA/LAMA | 10 (33.3) | 15 (50.0) | 25 (41.7) |
| ICS | 2 (6.7) | 2 (6.7) | 4 (6.7) |
| ICS/LABA | 15 (50.0) | 17 (56.7) | 32 (53.3) |
| Steroids | 2 (6.7) | 1 (3.3) | 3 (5.0) |
| Methylxanthine | 2 (6.7) | 2 (6.7) | 4 (6.7) |

**Table 1** Continued

| Characteristic | Intervention n=30 | Usual care n=30 | All n=60 |
|---|---|---|---|
| Other | 9 (30.0) | 9 (30.0) | 18 (30.0) |
| None of the above | 6 (20.0) | 4 (13.3) | 10 (16.7) |
| Smoking status n (%) | | | |
| Current | 10 (23.3) | 13 (43.3) | 23 (38.3) |
| Ex-smoker | 22 (73.4) | 21 (70.0) | 43 (71.7) |
| Never smoker | 2 (6.67) | 2 (6.7) | 4 (6.67) |
| GOLD stage, n (%) | | | |
| IV (FEV$_1$ <30% predicted) | 1 (3.3) | 1 (3.3) | 2 (3.3) |
| III (FEV$_1$ 30%–49% predicted) | 4 (13.3) | 4 (13.3) | 8 (13.3) |
| II (FEV$_1$ 50%–79% predicted) | 19 (63.3) | 18 (60.0) | 37 (61.7) |
| I (FEV$_1$ ≥80% predicted) | 6 (20.0) | 7 (23.3) | 13 (21.7) |
| MRC Dyspnoea Score, mean, (SD) | 3.4 (0.6) | 3.2 (0.4) | 3.3 (0.5) |
| MRC Score 3, n (%) | 21 (70.0) | 24 (80.0) | 45 (75.0) |
| MRC Score 4, n (%) | 7 (23.3) | 6 (20.0) | 13 (21.7) |
| MRC Score 5, n (%) | 2 (6.7) | 0 (0.0) | 2 (3.3) |
| COPD Assessment Test (CAT) Score, mean (SD) | 20.3 (5.9) | 19.9 (6.6) | 20.1 (6.2) |
| PHQ-9 Depression Score, mean (SD) | 6.2 (4.0) | 5.3 (4.1) | 5.8 (4.0) |
| GAD-7 Anxiety Score, mean (SD) | 3.5 (3.2) | 3.3 (2.8) | 3.4 (3.0) |
| Stanford Self-Efficacy Score, mean (SD) | 5.7 (1.9) | 6.7 (1.5) | 6.2 (1.7) |
| ISWT, mean (SD) | 229.7 (112.7) | 249.3 (96.2) | 239.5 (104.3) |
| Self-reported number of exacerbations in the last 6 months, mean (SD) | 2.7 (0.9) | 2.3 (1.1) | 2.5 (1.1) |
| St George's Respiratory Questionnaire (SGRQ) | | | |
| SGRQ—Impact, mean (SD) | 42.0 (19.9) | 37.0 (20.3) | 39.5 (20.1) |
| SGRQ—Activity, mean (SD) | 67.0 (20.2) | 56.9 (24.1) | 62.0 (22.6) |
| SGRQ—Symptoms, mean (SD) | 64.2 (17.1) | 65.4 (15.0) | 64.8 (16.0) |
| SGRQ—Total, mean (SD) | 53.6 (17.2) | 48.1 (17.9) | 50.9 (17.6) |
| Total Physical Activity Scores, MET-minutes/week (SD) | 2092.8 (2647.2) | 3027.7 (3303.9) | 2560.3 (3005.3) |

COPD, chronic obstructive pulmonary disease; FEV$_1$, forced expiratory volume in 1 s; GAD-7, Generalised Anxiety Disorder-7; GOLD, Global Initiative for Chronic Obstructive Lung Disease; ICS, inhaled corticosteroid; ISWT, incremental shuttle walk test; LABA, long-acting beta agonist; LAMA, long-acting muscarinic antagonist; MET, metabolic equivalent of task; PHQ-9, Patient Health Questionnaire-9; SABA, short-acting beta agonist; SAMA, short-acting muscarinic antagonist.

## Completeness of follow-up questionnaires

The level of completion of the questionnaire was high in all cases where the questionnaire was completed at the assessment and over the phone with 100% completion of questions.

## Feasibility of economic evaluation in definitive RCT

The EQ-5D-5L and self-reported COPD-related health-care utilisation questions had high levels of completion. Intervention participants reported costs mainly related to transportation (bus, petrol fuel, taxi). The cost of the programme came up as an important determinant for attendance in the future. Most participants noted that they would not be able to attend PR if they had to pay for it. A small number noted they would be able to pay €15–20 or 10% of the fee. Equipment costs, number of sessions delivered and staff costs for PR session delivery were established; the component costs are detailed in online supplemental table S4.

## Outcomes of a future RCT

At programme end (8 weeks), those with follow-up data in the intervention group (n=13) showed substantial improvements across all the outcomes including the SGRQ and ISWT, with mean changes (SD) of −24.9 (25.4) (95% CI -40.3 to −9.6) and 120.8 m (89.5), respectively. The usual care group (n=25) showed only minimal changes (table 2). At the 6 months follow-up improvements in the intervention (n=27) were still visible, but much more modest. The usual care group showed considerable improvements at the 6 months follow-up for

**Table 2** Effect of pulmonary rehabilitation at 8 weeks and 6 months for patients in the intervention and usual care groups

| | 8 weeks follow-up | | | | 6 months follow-up | | | |
| | Intervention | | Usual care | | Intervention | | Usual care | |
| | Mean (SD) n=13 | Mean change from baseline (SD) (95% CI) | Mean (SD) n=25 | Mean change (SD) (95% CI) | Mean (SD) n=27 | Mean change from baseline (SD) (95% CI) | Mean (SD) n=26 | Mean change from baseline (SD) (95% CI) |
|---|---|---|---|---|---|---|---|---|
| SGRQ—Total* | 32.3 (27.4) | −24.9 (25.4) (−40.3 to −9.6) | 48.0 (20.8) | −0.5 (18.0) (−8.1 to 7.0) | 48.1 (23.5) | −4.4 (19.8) (−12.3 to 3.4) | 40.8 (20.0) | −8.1 (20.8) (−16.5 to 0.3) |
| SGRQ—Impacts | 23.3 (25.6) | −22.7 (22.6) (−36.4 to −9.1) | 37.6 (23.2) | 0.3 (22.9) (−9.2 to 9.7) | 38.4 (23.6) | −2.4 (21.0) (−10.7 to 5.9) | 30.1 (20.0) | −7.5 (25.5) (−17.7 to 2.9) |
| SGRQ—Activity | 42.5 (32.9) | −30.5 (29.4) (−48.3 to −12.8) | 58.6 (20.3) | 1.1 (17.7) (−6.2 to 8.5) | 58.5 (29.8) | −7.8 (26.5) (−18.3 to 2.7) | 49.2 (24.8) | −10.5 (23.3) (−19.9 to −1.1) |
| SGRQ—Symptoms | 41.0 (31.6) | −21.6 (37.3) (−44.1 to 1.0) | 60.5 (25.7) | −4.2 (21.4) (−13.2 to 4.9) | 58.1 (23.4) | −4.5 (24.1) (−14.1 to 5.0) | 57.7 (24.8) | −6.1 (22.4) (−15.1 to 3.1) |
| MRC Dyspnoea Score | 2.5 (1.2) | −0.9 (1.3) | 3.0 (0.6) | −0.2 (0.6) | 2.9 (1.0) | −0.4 (1.0) | 2.8 (0.9) | −0.3 (0.8) |
| COPD Assessment Test (CAT) Score† | 13.8 (8.5) | −6.6 (7.4) | 18.3 (7.0) | −1.4 (5.0) | 17.9 (8.5) | −1.7 (7.3) | 17.7 (7.6) | −2.7 (6.5) |
| PHQ-9 Depression Score‡ | 2.8 (1.9) | −2.8 (2.9) | 5.5 (5.1) | −0.3 (3.9) | 4.74 (4.2) | −1.2 (3.8) | 3.8 (3.0) | −1.9 (3.2) |
| GAD-7 Anxiety Score‡ | 1.9 (1.6) | −0.9 (3.0) | 3.3 (3.5) | −0.3 (2.8) | 2.51 (3.3) | −0.7 (2.8) | 2.0 (2.1) | −1.3 (2.6) |
| Stanford Self-Efficacy Score | 7.8 (1.2) | 1.9 (1.5) | 6.9 (1.5) | 0.3 (1.8) | 7.0 (1.5) | 1.2 (2.1) | 7.3 (1.2) | 0.7 (1.5) |
| ISWT§ | 316.2 (134.3) | 120.8 (89.5) | 247.6 (113.5) | −13.2 (110.7) | 217.6 (89.6) | −7.1 (92.5) | 216 (99.4) | −29.2 (74.1) |
| Number of exacerbations in the last 6 months | 3.5 (1.1) | 0.7 (1.3) | 3.6 (1.0) | 1.3 (1.7) | 3.2 (1.3) | 0.5 (2.5) | 3.2 (1.3) | 0.92 (1.7) |
| Total Physical Activity Scores. MET-minutes/ week (SD) | 783.2 (1509.9) | −1309.6 (2539.0) | 1634.7 (1532.0) | −1393.1 (2838.3) | 943.6 (1163.5) | −1653.3 (4650.6) | 1857.0 (2853.4) | −2386.7 (5537.9) |

*A reduction in SGRQ is a positive patient outcome, minimal clinically important difference (MCID) 4 points.
†MCID: −2 points.
‡MCID 20% reduction.
§MCID 35–36 m
COPD, chronic obstructive pulmonary disease; GAD-7, Generalised Anxiety Disorder-7; ISWT, incremental shuttle walk test; MET, metabolic equivalent of task; PHQ-9, Patient Health Questionnaire-9; SGRQ, St George's Respiratory Questionnaire.

**Table 3** Self-reported tobacco use at 8 weeks and 6 months

| | 8 week follow-up | | 6 month follow-up | |
|---|---|---|---|---|
| **Tobacco use** | **Intervention n=13 n (%)** | **Usual care n=25 n (%)** | **Intervention n=27 n (%)** | **Usual care n=26 n (%)** |
| Current smoker at study start | 3 (23.1%) | 10 (40.0%) | 9 (33.3%) | 9 (34.6%) |
| Current smoker at follow-up | 3 (23.1%) | 8 (32.0%) | 10 (37.0%) | 7 (26.9%) |
| Smokers at baseline who tried to quit since enrolling in the research study | 2 (66.7%) | 3 (30.0%) | 4 (44.4%) | 4 (44.4%) |
| Smokers at baseline who quit since enrolling in the trial (cotinine validated) | 0 (0%) | 1 (10.0%) | 1 (11.1%) | 2 (22.2%) |

all the outcomes except for the ISWT and exacerbations (table 2).

Outcomes in relation to smoking are reported in table 3. At the 8 weeks follow-up none of the participants had quit smoking, but two in the intervention and three in the usual care groups had made quit attempts. By the 6 months follow-up, one in the intervention group and two in the usual care groups had quit smoking and an additional four in each group had made quit attempts.

### Attendance and adherence

Eight participants attended all 16 sessions, 13 (43.3%; or 54.2% of those who attended at least one session) attended at least 75% and 6 did not attend any session (online supplemental figure S1). Participants gave several

reasons for not attending PR sessions particularly highlighting 'cold weather, family issues and not being in town'.

### Per-protocol analysis

At the 6 months assessment only 13 intervention participants who had completed eight (50%) or more PR sessions were followed up. Table 4 reports the perprotocol analysis at 6 months; the 8 weeks results are in online supplemental table S5.

The SGRQ would be the primary outcome of a future RCT and at 6 months large, sustained improvements were seen across all the domains for the intervention group. A consistent pattern of sustained improvements was seen

**Table 4** Post hoc per-protocol analysis: 6 months follow-up of intervention patients who attended 50% or more pulmonary rehabilitation sessions compared with usual care

| | Intervention n=13 | | Usual care n=26 | |
|---|---|---|---|---|
| | **Mean (SD)** | **Mean change (SD) (95% CI)** | **Mean (SD)** | **Mean change (SD) (95% CI)** |
| SGRQ—Total* | 44.9 (24.3) | −16.6 (14.5) (−25.3 to 7.8) | 40.8 (20.0) | −8.1 (20.8) (−16.5 to 0.3) |
| SGRQ—Impacts | 36.6 (22.7) | −14.1 (16.8) (−24.3 to -4.0) | 30.1 (20.0) | −7.5 (25.5) (−17.7 to 2.9) |
| SGRQ—Activity | 53.3 (33.7) | −22.9 (21.3) (−35.8 to -10.1) | 49.2 (24.8) | −10.5 (23.3) (−19.9 to -1.1) |
| SGRQ—Symptoms | 54.7 (24.8) | −12.6 (20.1) (−24.7 to -0.4) | 57.7 (24.8) | −6.0 (22.4) (15.1 to 3.1) |
| MRC Dyspnoea Score | 2.6 (1.0) | −1.0 (0.8) | 2.8 (0.9) | −0.3 (0.8) |
| CAT† | 16.2 (7.1) | −5.5 (4.8) | 17.7 (7.6) | −2.7 (6.5) |
| PHQ-9 Depression Score‡ | 4.8 (5.0) | −0.9 (3.4) | 3.8 (3.0) | −1.9 (3.2) |
| GAD-7 Anxiety Score‡ | 2.8 (3.6) | −0.8 (3.2) | 2.0 (2.1) | −1.3 (2.6) |
| Stanford Self-Efficacy Score | 7.3 (1.7) | 1.6 (2.0) | 7.3 (1.2) | 0.7 (1.5) |
| ISWT§ | 217.5 (85.7) | 17.5 (72.6) | 216 (99.4) | −29.2 (74.1) |
| Self-reported number of exacerbations in the last 6 months | 2.9 (1.5) | 0.1 (1.5) | 3.2 (1.27) | 0.9 (1.7) |
| Total Physical Activity Scores, MET-minutes/week | 1078.5 (1498.5) | 72.92 (1816.8) | 1857.0 (2853.4) | −1170.7 (2753.3) |

*Minimal clinically important difference (MCID) 4 points.
†MCID −2 points.
‡MCID 20% reduction.
§MCID 35–36 m
CAT, COPD Assessment Test; COPD, chronic obstructive pulmonary disease; GAD-7, Generalised Anxiety Disorder-7; ISWT, incremental shuttle walk test; MET, metabolic equivalent of task; PHQ-9, Patient Health Questionnaire-9; SGRQ, St George's Respiratory Questionnaire.

across all the outcomes, apart from the ISWT which showed a modest improvement at 6 months.

## Intervention fidelity

Observation of one rehabilitation session and assessment of the rehabilitation specialists' records identified high levels of fidelity with the intervention delivery. In all participants there was an increase in numbers of repetitions/intensity of at least some of the exercises over their participation period.

"...their exercise intensity is increasing, number of sets is increasing" (Rehab specialist)

## Intervention acceptability

Most patients had a positive experience with the exercise, specialists, educational sessions and the booklet. Education sessions and guidance by the rehabilitation specialists were highly valued.

Most patients felt satisfied with the programme; some mentioned they felt it did not influence their behaviour much, while some emphasised that they felt more 'joyful'. Transport, hot weather during summer, health, money and work-related reasons were mentioned as barriers to continuing with the PR programme. Overall, the exercises were well received but suggestions to include more breathing exercises were made.

"(my physical activity) has changed, I walk more freely. When I need to lift heavy things, it is not difficult for me. It worked very well on me, and psychologically I am very well disposed" (P2)

Barriers: "weather and my health" (P4)

"I missed last two sessions … Transportation was a challenge" (P5)

The rehabilitation specialists reported that many participants improved and that attending the rehabilitation in a group increased their motivation.

"Those who completed 16 exercise sessions have asked us to print out exercise sheets. Some of the patients who had completed the exercises asked us to continue their exercises here, since the atmosphere here was different and group exercise was increasing their motivation" (Rehab specialist)

"Many patients did not know how to manage breathing, how to go outside confidently, did not know how to manage sexual relations, nutrition, what would be beneficial and what was not necessary. COPD patients usually are thin, because of their disease. Due to physical activity, one patient gained 7 kg and another one 5 kg, which made them very happy and they say result of the exercises" (Rehab specialist)

However, for some participants motivation was an issue as well as barriers of travel and time.

"but patients were missing sessions nevertheless. Those who had money, did not have time, and some did not work and transportation was a problem" (Rehab specialist)

## Adverse events

There were no adverse events reported throughout the study.

## DISCUSSION

This study aimed to determine the feasibility of delivering a culturally tailored PR programme in Georgia. Our results indicate that it was feasible to train rehabilitation specialists to deliver the PR with fidelity. There were challenges in recruiting participants from primary care, as many general practitioners do not have access to spirometry within the primary care setting. As a consequence, many patients assessed from primary care did not have an objective measure to support their diagnosis and on spirometry did not fulfil the criteria for COPD. Adherence to the programme was suboptimal with only 50% of intervention group participants attending at least 50% of the planned sessions. The programme was acceptable to the participants who attended, but clearly the fact that 20% of intervention participants did not attend any sessions suggests that there were considerable barriers to attendance. The rehabilitation specialists reported high dropout rates in participants who lived outside Tbilisi and had long journey times (<1 hour). Participants who attended demonstrated clinically significant improvements in outcomes, exceeding the minimally important clinical differences for the SGRQ, CAT, PHQ-9, GAD-7 and ISWT.[37–40] In a post hoc per-protocol analysis of intervention group participants who attended at least 50% of planned sessions, the outcomes were well sustained to 6 months.

The Cochrane review of PR for COPD[10] reported significant improvements in all the domains of the Chronic Respiratory Questionnaire and for the SGRQ, with the total score improving more than the minimally clinically important difference of 4 points, which was mirrored in our study. Of the 65 RCTs included the majority were from high-income settings and many of those from LMICs had small sample sizes.

We undertook some cultural tailoring of the PR before implementation, but did not find a need to make major changes to the programme given the delivery context in a secondary care setting. However, the discussion of end-of-life care was deemed unacceptable by participants from all three focus groups during the cultural adaptation, so was not included. In other studies, researchers have made significant adaptations to PR for patients who have recovered from tuberculosis, in Uganda, reporting high levels of engagement and good outcomes,[41] and in Greece,[33] and evaluated implementation in non-randomised studies. Several ongoing studies of PR in LMICs are registered on clinical trials registers.

## Strengths and limitations

This study has a number of strengths. While there have been many trials of PR, there are no published findings of trials undertaken in the Georgian setting. The topic was selected following a structured prioritisation process involving policy makers, clinicians and patients in Tbilisi, Georgia.[42] We undertook some cultural tailoring prior to finalising the final form of PR to be delivered to increase acceptability. The training was delivered by experienced trainers from the UK.

The study also has some limitations. A key issue was the low follow-up rate at the end of the programme in the intervention group resulting in an overall follow-up rate of 63% at 8 weeks, similar to results reported in other studies.[43 44] The differential follow-up at 8 weeks makes the findings difficult to interpret. The fall-off in effects between 8 weeks and 6 months in the intervention group might have been the result of more adherent participants attending the 8-week follow-up. To address this issue our post hoc per-protocol analysis explored the effects of the intervention in those participants most adherent to the PR programme and identified sustained improvements in health outcomes in this group. We faced challenges in recruiting primary care patients with COPD, largely due to the lack of availability of spirometry for diagnosis.[8] This could be overcome by alternatives to spirometry such as peak flow and questionnaires to diagnose COPD.[45] Studies from other settings have also reported structural and healthcare practitioner factors affecting the quality of care delivered for COPD.[46–48]

In our study only 43% would be categorised as having completed the programme, that is, attended at least 75% of the planned sessions. Adherence to PR is variable with reported non-completion rates from 42% across a range of high-income settings[49] to 38% in the UK in 2017.[50] Previous PR research has reported similar barriers to our study, including transportation difficulties, financial barriers, lack of support, illness and lack of motivation.[51] The literature reports large dropout rates between referral and preassessment and up to 15% of participants not continuing to PR following the preassessment.[50 52] We randomised participants prior to the prerehabilitation session and 20% of our intervention participants did not commence the PR programme; other researchers have reported high levels of non-attendance with up to 50% of people referred to PR not attending a single session.[53]

Our study findings may not apply to patients with COPD managed in primary care in Georgia, as 89% of our study participants were recruited from secondary care and therefore had more severe disease. If PR is implemented in Georgia, referrals will either need to be from secondary care, or spirometry or other diagnostic methods[45] included in the referral pathway from primary care, or inclusion criteria for PR widened to include those without confirmed COPD.

Our population may not be representative of all those with COPD in Georgia as a high proportion of participants were from secondary care, had considerable symptoms (all MRC Dyspnoea Score 3 or more), yet 16.7% were not taking medication for their lung condition. Our qualitative sample of respondents was relatively small, but the researchers felt that thematic saturation was achieved; the sample may not have been generalisable to all those who took part.

## Recommendations for future research

This study highlighted deficiencies in COPD diagnosis in primary care in Georgia and future research could explore ways to strengthen management of CRDs outside the secondary care setting.

A large trial or implementation research in the Georgian context would establish optimal delivery and effectiveness of PR for COPD. Additionally, the effectiveness of PR in Georgia for patients with other respiratory diseases and post-COVID lung damage might be undertaken.[54–56] The decision about progression to a full trial or an implementation study should draw on the feasibility outcomes of findings. Given the very minor adaptations made to the PR programme we would argue that future research should focus on best methods of implementation rather than effectiveness.

For future intervention delivery, the issue of accessibility needs to be considered, with rehabilitation provision near to where people live. The PR programme was delivered in a secondary care setting in Georgia and consideration should be given to delivery in community settings, or even home-based programmes. These could draw on the remote[57] and home-based programmes delivered in response to the COVID-19 pandemic.[58] Widespread roll-out would require training of the rehabilitation specialist workforce in Georgia. Another significant barrier is cost, as PR is not currently included within the free healthcare provision in Georgia. However, since this research was prioritised by policy makers in Georgia there is a clear recognition that it may be valuable, offering a possible route to change this in the future.

## CONCLUSIONS

These findings demonstrate the feasibility of PR delivery in Georgia and showed promising outcomes for those participants who adhered to the intervention. However, investment to enable primary care to diagnose and refer people with COPD is needed. Challenges with retention suggest that future PR programmes need to be delivered close to where people live and possibly provide flexibility in methods of delivery.

**Author affiliations**
[1]Georgian Respiratory Association, Tbilisi, Georgia
[2]Petre Shotadze Tbilisi Medical Academy, Tbilisi, Georgia
[3]Ivane Javakhishvili Tbilisi State University Faculty of Medicine, Tbilisi, Georgia
[4]Tbilisi State Medical University Faculty of Medicine, Tbilisi, Georgia
[5]M. Iashvili Children's Central Hospital, Tbilisi, Georgia
[6]Department of Physiotherapy, University Hospitals Birmingham NHS Foundation Trust, Birmingham, UK
[7]iCares Directorate, Sandwell and West Birmingham Hospitals NHS Trust, Birmingham, UK

[8]Institute of Applied Health Research, University of Birmingham College of Medical and Dental Sciences, Birmingham, UK

[9]Department of General Practice, Peking University First Hospital, Beijing, 100034, China

[10]Lung Function & Sleep, University Hospitals Birmingham NHS Foundation Trust, Birmingham, UK

[11]Life and Health Sciences Research Institute, University of Minho, Braga, Portugal

[12]International Primary Care Respiratory Group, Edinburgh, Scotland

[13]Observational and Pragmatic Research Institute, Singapore

[14]Health Services Management Centre, University of Birmingham College of Arts and Law, Birmingham, UK

[15]Family Medicine, ABC Medical School, São Paulo, Brazil

[16]NIHR Birmingham Biomedical Research Centre, University Hospitals Birmingham NHS Foundation Trust and University of Birmingham, Birmingham, UK

[17]Centre for Family Medicine, Medical Faculty, Skopje, North Macedonia

[18]Faculty of Medicine, University of São Paulo, São Paulo, Brazil

**Acknowledgements** The authors thank the International Primary Care Respiratory Group for introducing them to the primary care networks involved in this study and for its continued facilitation of clinical engagement. The authors also thank the Trial Steering Group Committee and the International Scientific Advisory Committee: Prof Niels Chavennes, Leiden University (Chair); Dr Semira Manaseki-Holland, University of Birmingham; Dr N Zhvania (Professor in Cardiology, Georgia); Revaz Vachnadze (patient representative) Georgia; Prof Debbie Jarvis, Imperial College London (Chair); Dr Nega Milevsla, Centre for Regional Policy Research and Cooperation 'Studiorum'; Skopje; Gordna Kunovska, Skopje. The authors especially thank: the patients included in the study, the Chapidze Emergency Cardiology Center for providing the rehabilitation facilities, Kelly Redden-Rowley and Helen Beadle for delivering the PR training, the Georgian rehabilitation specialists Shota Bzarashvili and Teona Kvitsadze for delivering the PR. The authors also thank Radmila Ristovska (1955–2020), who was involved in the initiation of this study.

**Contributors** KJ, RJ, PA co-led the study design, with contributions and advice from all other authors. TM was the local PI and oversaw all activities in Georgia. The cultural adaptation was overseen by NM, KJ, KR. The team from Georgia (MM, IK, NM, TM, IC, KG, NC) contributed to study design, day to day trial management, recruitment, intervention and follow up overseen by KJ, RA, KR, AT, RJ, PA. AE oversaw the quality of lung function testing. KR-R, HB delivered the PR training to the Georgian rehabilitation specialists. NM, MM collected the qualitative data overseen by KR, RA, NG. AS, SJ designed the analysis plan and economic data collection, respectively. IK conducted the statistical analysis, supported by AS, AD, KJ. KJ, MM, NM, IK wrote the manuscript with input from all other authors. As part of the Breathe Well Global Health Research Group, CC, KKC, BC, JC-d-S, AF, SM, KS, RS, SW contributed to the initial topic identification, protocol development and ongoing oversight of this study. All authors contributed to and approved the final version. KJ acts as guarantor for the study.

**Funding** This research was funded by the National Institute for Health Research (NIHR) (NIHR global group on global COPD in primary care, University of Birmingham, (project reference: 16/137/95) using UK aid from the UK Government to support global health research. KJ is part funded by the NIHR Applied Research Collaboration West Midlands (ARCWM). The views expressed in this publication are those of the author(s) and not necessarily those of the NIHR, ARCWM or the UK Department of Health and Social Care.

**Competing interests** None declared.

**Patient and public involvement** Patients and/or the public were involved in the design, or conduct, or reporting, or dissemination plans of this research. Refer to the Methods section for further details.

**Patient consent for publication** Not applicable.

**Ethics approval** This study involves human participants and was approved by Acad. G.Chapidze Emergency Cardiology Center, Medical Ethical Committee (session protocol №2-2018) and by the University of Birmingham (ERN_18-0856).

**Provenance and peer review** Not commissioned; externally peer reviewed.

**Data availability statement** Data are available upon reasonable request.

**ORCID iDs**
Rachel Adams http://orcid.org/0000-0002-1798-3854
Andrew P Dickens http://orcid.org/0000-0002-7591-8129

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
