## [Reviewer comments · BMJ Open]

ARTICLE DETAILS

TITLE (PROVISIONAL)	Feasibility of a pulmonary rehabilitation programme for patients with symptomatic chronic obstructive pulmonary disease in Georgia: a single site, randomized controlled trial from the Breathe Well Group
AUTHORS	Maglakelidze, Mariam; Kurua, Ia; Maglakelidze, Nino; Maglakelidze, Tamaz; Chkhaidze, Ivane; Gogvadze, Ketevan; Chkhaidze, Natia; Beadle, Helen; Redden-Rowley, Kelly; Adab, Peymane; Adams, Rachel; Chi, Chunhua; Cheng, Kar Keung; Cooper, Brendan; Correia-de-Sousa, Jaime; Dickens, Andrew; Enocson, Alexandra; Farley, Amanda; Gale, Nicola K.; Jowett, Sue; Martins, Sonia; Rai, Kiran; Sitch, Alice; Stavrikj, Katarina; Stelmach, Rafael; Turner, Alice; Williams, Sian; Jordan, Rachel; Jolly, Kate

VERSION 1 – REVIEW

REVIEWER	Rupert Jones University of Plymouth
REVIEW RETURNED	01-Nov-2021

GENERAL COMMENTS	This is a good paper on an important subject- reporting a programme of research implementing and assessing cultural adapted pulmonary rehab in Georgia. As a pilot RCT in LMICs I think it may well be a first, if so, perhaps it should say so. The RCT had a good sample size and with many outcomes assessed over a long follow up period (6 months). For those completing the programme there were clear and surprisingly large improvements. It was a massive achievement to undertake this extensive programme of work during the pandemic. The paper is wide in scope reporting on (i) a comprehensive stakeholder engagement exercise including 3 focus groups (ii) a pilot feasibility RCT with 60 participants (iii) 9 semi-structured interviews of participants after completion of the intervention. It is well written and follows a structured logical approach with appropriate information at all stages. Some would argue that this was an implementation research project and could have been reported following the STaRi framework but I think the approach undertaken is fine. There are no serious concerns. The scope of the paper means that although vast amounts of information are provided (and clearly displayed), some valuable detail is lost especially from the qualitative elements and the economic evaluation. This research may have been better split in to 2 or more papers. I have listed some lesser comments and suggestions. Strengths / limitations
--

	“A post-hoc per-protocol analysis explored the outcomes in intervention participants who attended at least 50% of the pulmonary rehabilitation sessions, showing a consistent pattern of improvements” This sentence could be clearer -clinically important improvements? what type of outcome? Introduction “It is currently the 4th leading cause of death worldwide^{3,4} COPD is third leading cause of death in the world (https://www.who.int/news-room/fact-sheets/detail/the-top-10-causes-of-death) and reference 3 is not relevant (it’s about COPD costs) and reference 4 contains a link which does not work. Reference 10 is not ideal as it does not mention pulmonary rehab or its effectiveness. The introduction does not discuss existing literature on PR in LMICs and what is known and not known about their feasibility and implementation, the only reference is to a systematic review of controlled trials. If space allows, could the fact that Georgia is a middle-income country be mentioned? Intervention “The intervention was adapted from PR programmes delivered in the UK.” Why not consider culturally adapted rehab undertaken in LMICs? Later it states it was based on international guidance- UK and ERS/ATS. The intervention was delivered by 2 rehab specialists- were they physios, doctors or nurses- was this a multidisciplinary team? Probably because of space there is minimal description of the exercise component and its prescription in the text, more information is provided in the appendices, perhaps these could be linked? Was the delivery of PR affected by the pandemic? What was the group size? Did risks of Covid-19 affect recruitment / delivery? Outcome measures There were a lot of questionnaires- did this create an issue, were these all necessary for a future definitive RCT? Were these outcome measures assessed by people blinded to allocation? Is this feasible in a future RCT? Process evaluation 9 semi-structured interviews of participants after completion of the intervention after participation- I am not convinced that 9 is sufficient or that saturation occurred. Results The majority of participants did not receive optimal prescribed medicine according to international guidelines when they entered the trial- this creates 2 issues-  1. It is usual practice to ensure that prior to starting PR patients are on optimal medical therapy. 2. Did the assessment process for the trial lead to changes in
--	--

	medication being prescribed to participants? If so this could have confounded the improvement from rehab. The improvements in SGRQ and ISWT were startling. The SGRQ improved by 24.9 points, the MCID is only 4 and in the definitive 2015 Cochrane meta-analysis on PR, the mean improvement was only 7. The ISWT improved by 120 m, whereas in the Cochrane meta-analysis it was 40m. How come that 26/30 of the usual care group attended for assessment at 8 weeks and only 13/30 in the intervention group? What conclusion can be made about the feasibility of a usual care group attending and completing outcome measures in a future fully powered RCT, despite no active intervention. Were the usual care group offered rehab later? Ethically some suggest waiting list controls are more appropriate. The Number of exacerbations in the last 6 months is reported in an 8 week period- how was this estimated? (table 2) The results section necessarily omitted a lot of information from focus groups and interviews which is a shame, but inevitable given the scope of the manuscript. The discussion is strong. The strength and limitations section is duplicated. It may be relevant to indicate in the future studies section that trials.gov has many RCTs of pulmonary rehab registered in LMICs. The tables and figures are generally good, although I did feel some of the appendices may not be needed. In tables 2,4 and S5 inclusion of the MCIDs of the tools may help the reader to interpret the results. Overall This is a very comprehensive report on an important series of studies in one paper. The quality of the paper is very good, but could benefit from some revisions as listed above. The references need to be carefully checked throughout, with so many authors I find it amazing that basic errors in the references were not detected. The paper, including supplements, runs to 75 pages and may have merited splitting into 2 or 3 papers.
--	---

REVIEWER	Pervin Ekren, Ege University Faculty of Medicine, Chest Diseases Department
REVIEW RETURNED	07-Nov-2021

GENERAL COMMENTS	The authors planed the study protocol clearly and in detail. The article was written perfectly. But these results is not novel on the world. My suggestion is that the author can apply to local journal to publish.
--

REVIEWER	Fabrizio Minervini McMaster University, Department of Thoracic Surgery
REVIEW RETURNED	31-Jan-2022

GENERAL COMMENTS	Dear Authors, thanks for submitting your valuable paper. This feasibility trial is very interesting and even if is limited to a geographic region opens the doors for a larger trial that could include more patients and with a better follow up rate. I have no major revisions to suggest.
--

REVIEWER	Fanuel Bickton Malawi-Liverpool-Wellcome Trust Clinical Research Programme, Lung Health Research Group
REVIEW RETURNED	27-Feb-2022

GENERAL COMMENTS	SECTION A: FORMAT REVIEW There should be no abbreviations in the title (i.e., remove PR and RCT) and write words in full (e.g., COPD). The title page should have included the word count of the main text (i.e., excluding title page, abstract, references, figures, tables, acknowledgments, and other supplementary statements). Authors should have included keywords at the end of the abstract, relevant of the content of the manuscript. The bullet points under the “strengths and limitations” after the abstract would be written more concisely (i.e., not more than one line each). Format the references as per the journal guidelines e.g.: Authors should revise reference numbering order in the body of the manuscript – for example, in the “strengths and limitations” section, authors jump from reference superscript #45 to reference superscript #47. Where is reference superscript #46? Also, the reference superscript number at the of the third paragraph under this section is indicated as 456, which is awkward it’s too large – use of a reference software manager would address this. Use the approved NLM title abbreviations for journal names in your references, e.g., in your reference #24, you wrote “ECP” as an abbreviation for “Effective clinical practice” while the approved NLM title abbreviation is “Eff Clin Pract” (see at https://www.ncbi.nlm.nih.gov/nlmcatalog?term=%22Eff+Clin+Pract%22%5BTitle+Abbreviation%5D) Some journal references were not properly formatted as per journal guidelines, e.g., remove dates of access and include DOI rather than URL, e.g., I would write reference #27 as “Herdman M, Gudex C, Lloyd A, Janssen M, Kind P, Parkin D, et al. Development and preliminary testing of the new five-level version of EQ-5D (EQ-5D-5L). Qual Life Res. 2011;20(10):1727-36. DOI: 10.1007/s11136-011-9903-x” NOT “Herdman M, Gudex C, Lloyd A, et al. Development and preliminary testing of the new five-level version of EQ-5D (EQ-5D-5L). Qual Life Res 2011 Dec;20(10):1727–36. https://www.ncbi.nlm.nih.gov/pmc/articles/PMC3220807/ (date accessed 23/11/20).” Having said that, fix all references with similar issues. Unless the journal is not listed in Medline, use NLM title abbreviations when writing journal names, not their full names, e.g., “Journal of Evaluation in Clinical Practice” in your reference #12 should be abbreviated “J Eval Clin Pract” (see https://www.ncbi.nlm.nih.gov/nlmcatalog?term=%22J+Eval+Clin+Pract%22%5BTitle+Abbreviation%5D). Also, this reference #12 has a similar format issue described in the previous bullet point for reference # 27 – fix these. Check journal guidelines on the correct format for references that are webpages (I usually refer to ICMJE sample of formatted references at https://www.nlm.nih.gov/bsd/uniform_requirements.html since BMJ
---

did not specify at <https://authors.bmj.com/writing-and-formatting/formatting-your-paper/>).

Authorship contributions:

The initials of author names used in this section should be for ALL NAMES of those authors as they appear in the byline on the title page, e.g., if the initials “RJ” belong to “Rachel E Jordan”, then the initials should be “REJ”; if the initials “AD” belong to “Andrew P Dickens”, the initials should be “APD”. Also, what does the question mark (?) on “NM” mean where you wrote “KJ, MM, NM?, IK wrote the manuscript...”? Also, there is a hyphen in “Kelly Redden-Rowley” – the initials should include a hyphen as “KR-R”, right? Also, put “and” before last author initials not comma, e.g., “NM and MM collected the qual...”; “KJ, RJ and PA co-led...” etc.

The corresponding author should ensure all individuals listed as co-authors met the ICMJE authorship criteria satisfactorily (<https://www.bmj.com/about-bmj/resources-authors/article-submission/authorship-contributorship>) and those that contributed but didn’t meet these criteria satisfactorily should be shifted to the acknowledgments section. The contributions of some authors are not specific and therefore questionable. Their contributions to the authorship of this specific manuscript should be specified, or they appear to be “honorary/guest/gift authors” to me, which would be scientific misconduct – the inclusion of a co-author whose contributions did not warrant authorship (check Rajasekaran S, Li Pi, Shan R, Finnoff JT. Honorary authorship: frequency and associated factors in physical medicine and rehabilitation research articles. Arch Phys Med Rehabil. 2014;95:418–28. <https://pubmed.ncbi.nlm.nih.gov/24215989/>). For example,

contributions of the following individuals to the authorship of this manuscript were not specified: Chunhua Chi (CC), Kar Keung Cheng (KJC), Brendan G Cooper (BGC), Jaime Correia-de-Sousa (JCS), Amanda Farley (AF), Sonia M Martins (SMM), Katarina Stavrikj (KS), Rafael Stelmach (RS), and Siân Williams (SW). If these were included as authors simply because they are “part of the Breathe Well Global Health Research Group”, as listed at <https://www.birmingham.ac.uk/research/applied-health/research/breathe-well/our-partners.aspx>, then I would argue (the Editor-in-Chief would decide, of course) that they do not qualify as authors for this specific manuscript, despite the fact that the manuscript is a product of the broad Breathe Well Group of which they are partners/collaborators. In that case, I would not include them as authors, but rather acknowledge them as collaborators/partners at the end of the manuscript, as well as include the statement “on behalf of the Breathe Well Group/partners/collaborators” at the end of the by-line of eligible authors on the title page. In fact, the authors of the current study did this well in their IPCRG conference abstract and presentation at <https://www.ipcr.org/11641> in which my suspected honorary/guest/gift authors mentioned above were rightly not included as authors and the statement “on behalf of the Breathe Well Group” was added at the end of the eligible authors’ by-line.

As another example, the authors may also refer to “Brakema EA, van der Kleij RMJJ, Poot CC, An PL, Anastasaki M, Crone MR, Hong LHTC, Kirenga B, Lionis C, Mademilov M, Numans ME, Oanh LTT, Tsiligianni I, Sooronbaev T, Walusimbi S, Williams S, Chavannes NH, Reis R; FRESH AIR collaborators. Mapping low-resource contexts to prepare for lung health interventions in four countries (FRESH AIR): a mixed-method study. Lancet Glob Health.

2022 Jan;10(1):e63-e76. doi: 10.1016/S2214-109X(21)00456-3. PMID: 34919858. <https://pubmed.ncbi.nlm.nih.gov/34919858/>” on how they did it.

Competing interests

The statement only declares competing interests for “the principal investigators” who, according to information in the manuscript’s “Contributors” section, is Tamaz Maglakelidze TM as this is the only author who was explicitly mentioned as the “local PI”. So, what about any competing interest of the other authors? I don’t think disclosure of competing interests should be restricted to PIs only. As the per BMJ guidelines, see the BMJ Author Hub at <https://authors.bmj.com/policies/competing-interests/> for details on what to include as competing interests.

Patient consent for publication: Not required.

Why was it not required, especially knowing that if this manuscript is published, it will include a publication of patient clinical data and interview quotes? BMJ recommends participants’ consent for both participation and publication (check <https://www.bmj.com/company/researchintegrity/consent-for-publication/>).

Data availability statement: All data are available on request from the corresponding author subject to a data sharing agreement. Are you referring to “raw” datasets (e.g., interview transcripts, etc)? Your manuscript also includes data (participants quotes, etc), no?

SECTION B: CONTENT REVIEW

This study was single-centred (not nation-wide), so the title should appropriately reflect this – “Georgia” is a country and, in the title, implies it was a nationally-representative study. The title should have specified the setting as “Chapidze Emergency Cardiology Center, Tbilisi, Georgia” which would make it easy for systematic reviewers of studies done in specific settings.

Introduction:

I think we (me included as a respiratory researcher) have been over-citing the statistic that “more than 90% of COPD-related deaths occur in LMICs” without properly tracing its original source. I would be wrong and I ask the authors for their thoughts, but I think it is a statistic that is somehow outdated or questionable or both – it is based on a 2010 report by Professor Ala Alwan, entitled “The Global Status Report on Noncommunicable Diseases” published on the WHO website at https://www.who.int/nmh/publications/ncd_report_full_en.pdf (so you may have to cite this original reference, not the one you cited in reference #6 by McCarthy et al, 2015). I did not have the chance to look at and critically appraise the research methods that were used to derive this statistic as the report only reports findings, not methods – hence, I find the statistic questionable. I encourage the authors to look at that original report. Finally, the authors should look at the more up-to-date report, Global Burden of Disease Study 2017 at [https://doi.org/10.1016/S2213-2600\(20\)30105-3](https://doi.org/10.1016/S2213-2600(20)30105-3), which suggests that the prevalence is higher in high-income countries, i.e.: “The global prevalence in 2017 was around 7.1% (95% UI 6.6–7.7). Chronic respiratory diseases were most prevalent across the GBD high-income super-region, at 10.6% (9.9–11.3), up from 9.7% (9.1–10.3) in 1990 (table 1). By contrast, the lowest prevalence was observed in sub-Saharan Africa (5.1% [4.5–5.8]) and south Asia

(5.5% [5.1–6.0]). Latin America and the Caribbean had the largest decline (–0.80 percentage points) in chronic respiratory disease prevalence over the study period, from 8.1% in 1990 to 7.3% in 2017. Sub-Saharan Africa and the central Europe, eastern Europe, and central Asia super-region also saw declines in the prevalence of chronic respiratory diseases.” My assumption is that the higher the prevalence, the higher the burden. Of course, COPD burden includes its related “deaths” but we just need to be sure of the quality of the 2010 research methods that established, as a fact, that “more than 90% of COPD-related deaths occur in LMICs”. I mean, this is 2022, a decade has gone.

Authors should provide references for the following statements:
“The availability of pharmacotherapy is limited due to resource constraints, making pharmacological management difficult.”
“In addition to the direct health and healthcare burden, COPD negatively impacts the health-related quality of life (HRQoL) of people living with the condition”

Authors should cite peer-reviewed journal publication(s), not an institutional website, for the statement “There is growing evidence that pulmonary rehabilitation (PR) is an effective and cost-effective therapeutic intervention to improve COPD symptoms, patients’ quality of life and to reduce risk of premature deaths.”

The statement that “there are currently no PR services offered to patients in Georgia” may not be entirely true because I saw webpages of some Georgian hospitals that advertise PR services (e.g., <https://clinchmh.org/service/pulmonary-rehabilitation/> and <https://breathingbetter.org/our-services/pulmonary-rehabilitation/>). Should the authors re-phrase their statement to indicate that there are few pockets of PR in Georgia (few but not completely absent), or that no PR studies exist on Georgia?

Overall, the introduction is well written and concise. It is interesting that the authors also bring the subject of a “culturally tailored” PR and I wonder what they mean by that. What is “cultural tailoring”? They could add a few lines to the introduction on this, i.e., what’s wrong with current PR packages that necessitated the researcher to adapt/tailor traditional PR to the Georgian setting? Such information would serve as a rationale for their decision to tailor/adapt it to the Georgian setting. Authors may check the publication at <https://doi.org/10.12688/wellcomeopenres.17702.1> on how authors elsewhere justified the need for their culturally adapted PR (introduction’s third paragraph).

Methods:

Under “Study design and participants”, authors should add the word “dyspnoea” after MRC to indicate that their inclusion criterion “MRC score ≥ 2 ” was for dyspnoea and the MRC Dyspnoea Scale was used, without which the reader could confuse it with the score for muscular power assessment which also uses a type of MRC scale. Under “Recruitment and randomisation”, write “... patients with COPD ...” not “... COPD patients ...”

Reference #17 (APEASE criteria) is a book chapter and should be cited as such (i.e., Michie S, Atkins L, West R. The APEASE criteria for designing and evaluating interventions. In: The Behaviour Change Wheel: A Guide to Designing Interventions. London: Silverback Publishing; 2014) NOT

	Under “Intervention”, the authors write “The adapted PR programme drew on international guidance” and provide references 16 and 20 to support this, but reference 16 is not international as it all its authors are from the UK and the publication is specifically a British guideline. Under “intervention”, authors should describe the exercises prescribed in the PR programme (e.g., type: strength/endurance/flexibility, lower limb/upper limb, warmup/cool down, interval/continuous training, exercise names/examples, etc) Under “intervention” the authors write “The programme took place twice weekly” and I am just wondering if they participants were advised to perform a third unsupervised home-based session as has been done in other studies. Under “intervention”, what education topics were offered to the control group, and why were they given PR educational booklet? Under “Outcome measures”, authors write that “Data were collected from patients by the research team members at the end of the PR programme (eight weeks) and at six months”. Were data also not collected before the commencement of the programme (at week 0), and why was a 6-months follow-up assessment done while the program ended at week 8? Also in this section, mention the outcomes that were assessed, and the outcome measures used. Under “feasibility outcomes”, how did the authors define and assess “acceptability”? http://dx.doi.org/10.1186/s12913-017-2031-8 The authors proceed to a “Secondary outcomes” section without having specified in the preceding sections what “primary outcomes” were. Or were the “feasibility outcomes” the primary outcomes? If yes, indicate them as such using explicit words “primary outcomes” as you did with “secondary outcomes”, rather than making the reader guess. Under “Secondary outcomes”, authors are not clear on the stage of the study at which the listed outcomes were assessed. For example, they are repetitive of the SGRQ in the second and third lines. In the second line, they write “at six months” and in the third line, they repeat the SGRQ and include both eight weeks and six months – confusing! Also, they write in the third line “SGRQ at programme end” – what was your programme? Is it at eight weeks or six months which you have already mentioned in the same third line before the colon? Also, the authors still do not mention whether they had assessed all the secondary outcomes mentioned in this section at baseline (at week 0). Again, why were the assessments repeated at six months when, it seems to me, the programme ended at eight weeks? Was it to assess the long-term impact of PR? Or you delivered a maintenance programme after eight weeks? Under “process evaluation”, the authors write “The average interview duration was 35 minutes for the participants and 33 minutes for the rehabilitation specialists.” Were the rehabilitation specialists not treated as study participants (i.e., consent etc)? Under “Data analysis section”, authors should not simply write that data were analysed using STATA/IC 15.1.3 – also tell us about the specific statistical tests you conducted in your data in STATA. I think the qualitative data analysis procedures done should be more comprehensively described. Results It is surprising that the authors said they aimed to recruit 60 participants and the flow diagram (figure 1) directly leads to this sample size. Please clarify in the methods section if 60 was planned before recruitment, or it was the number of eligible participants you were left with during pre-PR assessments, or it was by coincidence
--	---

that what you planned before recruitment (60 patients) was the same sample size (60 patients during pre-PR eligibility assessments).

Authors wrote in the methods section that p values and 95% CIs were not reported but I see some 95% CIs being reported in the results section, e.g., "... with mean changes (SD) of -24.9 (25.4) [95%CI -40.3, -9.6] and 120.8m (89.5) respectively."

As already commented elsewhere above, the authors did not tell us what pre-PR assessments were done at baseline (week zero). Those baseline/week zero measurements would have been included in Table 2, but they are not included in this table; hence the "mean changes" displayed in table 2 do not make sense since the reader/reviewer is not provided with baseline measurements with which to compare with measurements at 8 weeks and 6 months. Please display baseline measurements in table 2. Check the other tables too.

As an afterthought, were biometrics such as weight and BMI of participants measured? Would have been included in table 1 if important, especially also because I see one rehab specialists in the qualitative section reporting patients' weight gains in kilograms. Would this be an important outcome?

Outcome changes in the results section should have been reported against their minimal clinically important differences (MCIDs) available in literature (for ISWT distance, CAT, etc) since statistical significance tests couldn't be carried out.

I think the authors definition of "intervention fidelity" is too narrow. Is it really just about "increasing exercise prescriptions"? (Siedlecki, Sandra L. PhD, RN Research Intervention Fidelity, Clinical Nurse Specialist: 1/2 2018 - Volume 32 - Issue 1 - p 12-14 doi: 10.1097/NUR.0000000000000342. https://journals.lww.com/cns-journal/Citation/2018/01000/Research_Intervention_Fidelity__Tips_to_Improve.4.aspx#:~:text=Intervention%20fidelity%20means%20that%20the,internal%20validity%20of%20a%20study.)

I think that, due to journal word count limits, the authors haven't comprehensively reported data, especially the qualitative one (looking at the various questions contained in the interview topic guides vs the small number of participants' quotes in the results section – but I am more familiar with thematic analysis than content analysis, so the authors would know better). As a solution, other authors of a similar elsewhere split their data into two manuscripts (one for qualitative <https://doi.org/10.2147/COPD.S165623> and another for quantitative <https://doi.org/10.2147/copd.s146659>) and published these separately. But this would perhaps be considered as unethical by other editors (e.g., read about salami publication at <https://dx.doi.org/10.11613%2FBM.2013.030>) and I leave this to the BMJ Open editors to decide.

Discussion

As commented above, results should have been discussed in relation to published MCIDs where available (I know ISWT has an MCID, and I am sure the other outcome measures also have). The statement "Participants who attended demonstrated improvements in clinical outcomes" would confuse the reader as they wouldn't know whether you are talking of statistically significant improvement or clinically important improvements.

VERSION 1 – AUTHOR RESPONSE

Reviewer: 1

Dr. Rupert Jones, University of Plymouth Comments to the Author:

This is a good paper on an important subject- reporting a programme of research implementing and assessing cultural adapted pulmonary rehab in Georgia. As a pilot RCT in LMICs I think it may well be a first, if so, perhaps it should say so. The RCT had a good sample size and with many outcomes assessed over a long follow up period (6 months). For those completing the programme there were clear and surprisingly large improvements. It was a massive achievement to undertake this extensive programme of work during the pandemic.

Thank you

The paper is wide in scope reporting on (i) a comprehensive stakeholder engagement exercise including 3 focus groups (ii) a pilot feasibility RCT with 60 participants (iii) 9 semi-structured interviews of participants after completion of the intervention. It is well written and follows a structured logical approach with appropriate information at all stages. Some would argue that this was an implementation research project and could have been reported following the STaRi framework but I think the approach undertaken is fine.

Thank you.

There are no serious concerns. The scope of the paper means that although vast amounts of information are provided (and clearly displayed), some valuable detail is lost especially from the qualitative elements and the economic evaluation. This research may have been better split in to 2 or more papers. I have listed some lesser comments and suggestions.

Strengths / limitations

“A post-hoc per-protocol analysis explored the outcomes in intervention participants who attended at least 50% of the pulmonary rehabilitation sessions, showing a consistent pattern of improvements” This sentence could be clearer -clinically important improvements? what type of outcome?

We have added ‘in health related quality of life’.

Introduction

“It is currently the 4th leading cause of death worldwide^{3,4}” COPD is third leading cause of death in the world (<https://www.who.int/news-room/fact-sheets/detail/the-top-10-causes-of-death>) and reference 3 is not relevant (it’s about COPD costs) and reference 4 contains a link which does not work.

This has been amended to third leading cause of death worldwide and the reference corrected to the WHO fact sheet.

Reference 10 is not ideal as it does not mention pulmonary rehab or its effectiveness.

We have now referenced the Cochrane systematic review for the improvements in HRQoL and Rysø et al for the improvement in mortality.

The introduction does not discuss existing literature on PR in LMICs and what is known and not known about their feasibility and implementation, the only reference is to a systematic review of controlled trials.

Due to the tight word count we have kept this literature within the discussion section; including it in the introduction would lead to duplication.

If space allows, could the fact that Georgia is a middle-income country be mentioned?

We have added this to the introduction; this also clarifies that we are referring to the country Georgia, rather than the US State of Georgia.

There are currently no PR services offered to patients in Georgia, a middle income country.

Intervention

“The intervention was adapted from PR programmes delivered in the UK.” Why not consider culturally adapted rehab undertaken in LMICs? Later it states it was based on international guidance- UK and ERS/ATS.

Pulmonary rehabilitation has been culturally adapted for several different contexts and settings. We believed it would be best to start with a programme that adhered to international guidelines that were proven to improve health outcomes. Many adapted PR programmes have been shown to be implemented well, but many were not controlled studies and settings and contexts very different. We have improved the consistency of our description and referenced both sets of guidance:

The intervention was adapted from PR programmes delivered in the UK, and following UK and ERS/ATS guidance

The intervention was delivered by 2 rehab specialists- were they physios, doctors or nurses- was this a multidisciplinary team? Probably because of space there is minimal description of the exercise component and its prescription in the text, more information is provided in the appendices, perhaps these could be linked?

The rehabilitation was delivered by a cadre of health professionals known as rehabilitation specialists in Georgia, but similar to physiotherapists in the UK. They were supported by a team of medical professionals. We have added this detail to the text.

The intervention was delivered by two rehabilitation specialists (physiotherapists)

Was the delivery of PR affected by the pandemic? What was the group size? Did risks of Covid-19 affect recruitment / delivery?

No, delivery was completed just pre-pandemic.

Outcome measures

There were a lot of questionnaires- did this create an issue, were these all necessary for a future definitive RCT?

There wasn't an issue with the questionnaires. We have reported the high level of questionnaire completion in the results 'Completeness of follow-up questionnaires', so we do not plan to reduce the questionnaires should a full RCT be undertaken in this setting.

Were these outcome measures assessed by people blinded to allocation? Is this feasible in a future RCT?

Follow-up wasn't blinded. It would be feasible in a full RCT, but would need additional resourcing, not available in a feasibility study. We have clarified that the follow-up was not blinded in the methods. Add in the exact text.

Process evaluation

9 semi-structured interviews of participants after completion of the intervention after participation- I am not convinced that 9 is sufficient or that saturation occurred.

The qualitative research was analysed by content analysis. We have added some additional text to the discussion to address the issue of saturation and also generalisability:

Our qualitative sample of respondents was relatively small, but the researchers felt that thematic saturation was achieved; the sample may not have been generalizable to all those who took part.

Results

The majority of participants did not receive optimal prescribed medicine according to international guidelines when they entered the trial- this creates 2 issues-

1. It is usual practice to ensure that prior to starting PR patients are on optimal medical therapy.

2. Did the assessment process for the trial lead to changes in medication being prescribed to participants? If so this could have confounded the improvement from rehab.

The assessment process did not lead to changes in medication. The team were unable to deliver 'optimised medical therapy' prior to PR as this is not available to all patients in Georgia. We have added a sentence to clarify this in the methods:

Changes were not made to patients' medication.

The improvements in SGRQ and ISWT were startling. The SGRQ improved by 24.9 points, the MCID is only 4 and in the definitive 2015 Cochrane meta-analysis on PR, the mean improvement was only 7. The ISWT improved by 120 m, whereas in the Cochrane meta-analysis it was 40m.

This is likely to partially be due to responder bias, as discussed in the limitations section of the discussion. It is also possible that larger effects are due to the effectiveness of PR in people who are not already on fully optimised therapy.

How come that 26/30 of the usual care group attended for assessment at 8 weeks and only 13/30 in the intervention group? What conclusion can be made about the feasibility of a usual care group attending and completing outcome measures in a future fully powered RCT, despite no active intervention. Were the usual care group offered rehab later? Ethically some suggest waiting list controls are more appropriate.

We conclude that some of the intervention group found the intervention burdensome; Tbilisi is a large urban centre and travel times were long for some participants. We address this in the discussion:

The programme was acceptable to the participants who attended, but clearly the fact that 20% of intervention participants did not attend any sessions suggests that there were considerable barriers to attendance. The rehabilitation specialists reported high drop-out rates in participants who lived outside Tbilisi and had long journey times.

Unfortunately, there was not sufficient resource to offer a full PR programme to usual care, which would have been optimal. Instead, usual care participants were offered an educational session and the written educational resource provided to the intervention group:

They were offered a delayed 1.5 - 2 hour educational session delivered by pulmonologists and specialists once the final (6-months) follow-up was complete. During this session, participants received a PR educational booklet.

The Number of exacerbations in the last 6 months is reported in an 8 week period- how was this estimated? (table 2)

This was based on self-report, it is described in the methods section: Secondary outcomes.
self-reported number of exacerbations

The results section necessarily omitted a lot of information from focus groups and interviews which is a shame, but inevitable given the scope of the manuscript.

We are keen to keep the paper with the integrated quantitative and qualitative findings, particularly with a relatively small qualitative sample in the process evaluation.

The discussion is strong. The strength and limitations section is duplicated.

We have reduced the duplicated aspects from the recommendations for future research – please see page 23.

It may be relevant to indicate in the future studies section that trials.gov has many RCTs of pulmonary rehab registered in LMICs.

This has now been added

Several ongoing studies of PR in LMICs are registered on clinical trials registers.

The tables and figures are generally good, although I did feel some of the appendices may not be needed.

We would be happy to remove any appendices considered unnecessary.

In tables 2,4 and S5 inclusion of the MCIDs of the tools may help the reader to interpret the results.

We have added the MCID for the SGRQ as footnotes to each of these tables.

^aMCID 4 points; ^bMCID: -2 points; ^cMCID 20% reduction; ^dMCID 35-36m

Overall

This is a very comprehensive report on an important series of studies in one paper. The quality of the paper is very good, but could benefit from some revisions as listed above. The references need to be carefully checked throughout, with so many authors I find it amazing

that basic errors in the references were not detected. The paper, including supplements, runs to 75 pages and may have merited splitting into 2 or 3 papers.

We apologise for reference issues – we had technical issues with the reference manager file due to home working. We have now used reference manager software.

Reviewer: 2

Dr. Pervin Ekren, Ege University Faculty of Medicine Comments to the Author:

The authors planned the study protocol clearly and in detail. The article was written perfectly. But these results is not novel on the world. My suggestion is that the author can apply to local journal to publish.

Thank you, we appreciate the positive feedback. Regarding the novelty of the research, this is the first research on pulmonary rehabilitation coming from the country of Georgia and included cultural tailoring, which we believe to be novel.

Reviewer: 3

Dr. Fabrizio Minervini, McMaster University Comments to the Author:

Dear Authors,

thanks for submitting your valuable paper. This feasibility trial is very interesting and even if is limited to a geographic region opens the doors for a larger trial that could include more patients and with a better follow up rate. I have no major revisions to suggest.

Thank you.

Reviewer: 4

Mr. Fanuel Bickton, Malawi-Liverpool-Wellcome Trust Clinical Research Programme, University College London Comments to the Author:

SECTION A: FORMAT REVIEW

There should be no abbreviations in the title (i.e., remove PR and RCT) and write words in full (e.g., COPD).

These are now in full

The title page should have included the word count of the main text (i.e., excluding title page, abstract, references, figures, tables, acknowledgments, and other supplementary statements).

The word count has now been included on the title page

Authors should have included keywords at the end of the abstract, relevant of the content of the manuscript.

The key words have been added after the abstract

The bullet points under the “strengths and limitations” after the abstract would be written more concisely (i.e., not more than one line each).

We have reviewed the points in this section and reduced words where this will not affect clarity.

- *This is the first published pulmonary rehabilitation trial undertaken in Georgia.*
- *The intervention was culturally tailored for a middle income country, having been selected through a structured prioritisation process involving policy makers, clinicians and patients.*
- *The 63% follow-up in the intervention group at 8-weeks affects interpretation.*
- *A post-hoc per-protocol analysis explored the outcomes in intervention participants who attended at least 50% of the pulmonary rehabilitation sessions, showing a consistent pattern of improvements in health related quality of life*
- *Recruitment through primary care proved challenging due to patients with COPD diagnosis from primary care not fulfilling diagnostic criteria on spirometry.*

Format the references as per the journal guidelines e.g.:

Authors should revise reference numbering order in the body of the manuscript

Apologies for reference issues – we had technical issues with the reference manager file due to home working. We have now used reference manager software.

Use the approved NLM title abbreviations for journal names in your references, e.g., in your reference #24, you wrote “ECP” as an abbreviation for “Effective clinical practice” while the approved NLM title abbreviation is “Eff Clin Pract”

The journal names are now all in correct abbreviated format

Some journal references were not properly formatted as per journal guidelines, e.g., remove dates of access and include DOI rather than URL, e.g., I would write reference #27 as “Herdman M, Gudex C, Lloyd A, Janssen M, Kind P, Parkin D, et al. Development and preliminary testing of the new five-level version of EQ-5D (EQ-5D-5L). Qual Life Res. 2011;20(10):1727-36. DOI: 10.1007/s11136-011-9903-x”

Thank you, we have followed the journal’s guidance to “List the names and initials of all authors if there are 3 or fewer; otherwise list the first 3 and add ‘et al.’” URLs and access date for electronically accessible journals have been removed and DOIs added.

Unless the journal is not listed in Medline, use NLM title abbreviations when writing journal names, not their full names, e.g., “Journal of Evaluation in Clinical Practice” in your reference #12 should be abbreviated “J Eval Clin Pract”

This has been completed.

Also, this reference #12 has a similar format issue described in the previous bullet point for reference # 27 – fix these.

Thank you, the journal names are now all in correct abbreviated format

Check journal guidelines on the correct format for references that are webpages (I usually refer to ICMJE sample of formatted references at https://www.nlm.nih.gov/bsd/uniform_requirements.html since BMJ did not specify at <https://authors.bmj.com/writing-and-formatting/formatting-your-paper/>).

We have used the following guidance from the journal:

“Websites are referenced with their URL and access date, and as much other information as is available. Access date is important as websites can be updated and URLs change. The “date accessed” can be later than the acceptance date of the paper, and it can be just the month accessed”

Authorship contributions: The initials of author names used in this section should be for ALL NAMES of those authors as they appear in the byline on the title page.

We have corrected the author names and one minor typographical error.

The corresponding author should ensure all individuals listed as co-authors met the ICMJE authorship criteria satisfactorily (<https://www.bmj.com/about-bmj/resources-authors/article-submission/authorship-contributorship>) and those that contributed but didn’t meet these criteria satisfactorily should be shifted to the acknowledgments section. The contributions of some authors are not specific and therefore questionable. Their contributions to the authorship of this specific manuscript should be specified, or they appear to be “honorary/guest/gift authors” to me, which would be scientific misconduct. For example, contributions of the following individuals to the authorship of this manuscript were not specified: Chunhua Chi (CC), Kar Keung Cheng (KKC), Brendan G Cooper (BGC), Jaime Correia-de-Sousa (JCS), Amanda Farley (AF), Sonia M Martins (SMM), Katarina Stavrikj (KS), Rafael Stelmach (RS), and Siân Williams (SW). If these were included as authors simply because they are “part of the Breathe Well Global Health Research Group”, as listed at <https://www.birmingham.ac.uk/research/applied-health/research/breathe-well/our-partners.aspx>, then I would argue (the Editor-in-Chief would decide, of course) that they do not qualify as authors for this specific manuscript, despite the fact that the manuscript is a product of the broad Breathe Well Group of which they are partners/collaborators. In that case, I would not include them as authors, but rather acknowledge them as collaborators/partners at the end of the manuscript, as well as include the statement “on behalf of the Breathe Well Group/partners/collaborators” at the end of the by-line of eligible authors on the title page.

The Breathewell researchers worked as a collaborative group contributing to the projects in all the countries, this was because one explicit outcome was capacity building. The investigators from all the

countries were part of the process that identified the initial topics for investigation, they commented on and contributed to all the protocols and attended quarterly meetings to share solutions for ongoing problems and provide project oversight. All the named authors provided feedback on the draft manuscripts, in many cases quite extensively. We have amended the contributor section to make their contributions clearer:

As part of the Breathe Well Global Health Research Group, CC, KKC, BGC, JCS, AF, SMM, KS, RS, SS contributed to the initial topic identification, protocol development and ongoing oversight of this study. All authors contributed to and approved the final version.

Competing interests

The statement only declares competing interests for “the principal investigators” who, according to information in the manuscript’s “Contributors” section, is Tamaz Maglakelidze TM as this is the only author who was explicitly mentioned as the “local PI”. So, what about any competing interest of the other authors? I don’t think disclosure of competing interests should be restricted to PIs only. As the per BMJ guidelines, see the BMJ Author Hub at <https://authors.bmj.com/policies/competing-interests/> for details on what to include as competing interests.

The statement referred to all the investigators; we have clarified this to “none declared”

Patient consent for publication: Not required.

Why was it not required, especially knowing that if this manuscript is published, it will include a publication of patient clinical data and interview quotes? BMJ recommends participants’ consent for both participation and publication (check <https://www.bmj.com/company/researchintegrity/consent-for-publication/>).

Thank you, the guidelines state “A patient consent form: any article that contains personal medical information about an identifiable living individual requires the patient’s explicit consent before we can publish it. We will need the patient to sign our consent form, which requires the patient to have read the article. This form is available in multiple languages.”

https://bmjopen.bmj.com/pages/authors/#submission_guidelines

All the data here is anonymised, the above guidance only refers to identifiable data. As stated in the methods the participants all provided written informed consent prior to participation in the study.

Data availability statement: All data are available on request from the corresponding author subject to a data sharing agreement.

Are you referring to “raw” datasets (e.g., interview transcripts, etc)? Your manuscript also includes data (participants quotes, etc), no?

Yes, raw datasets.

SECTION B: CONTENT REVIEW

This study was single-centred (not nation-wide), so the title should appropriately reflect this – “Georgia” is a country and, in the title, implies it was a nationally-representative study. The title should have specified the setting as “Chapidze Emergency Cardiology Center, Tbilisi, Georgia” which would make it easy for systematic reviewers of studies done in specific settings.

This would make the title rather unwieldy so we have inserted ‘single site’ to clarify this:

Feasibility of a pulmonary rehabilitation programme for patients with symptomatic chronic obstructive pulmonary disease in Georgia: a single site, randomized controlled trial from the Breathe Well Group

Introduction:

I think we (me included as a respiratory researcher) have been over-citing the statistic that “more than 90% of COPD-related deaths occur in LMICs” without properly tracing its original source. I would be wrong and I ask the authors for their thoughts, but I think it is a statistic that is somehow outdated or questionable or both – it is based on a 2010 report by Professor Ala Alwan, entitled “The Global Status Report on Noncommunicable Diseases” published on

the WHO website at https://www.who.int/nmh/publications/ncd_report_full_en.pdf (so you may have to cite this original reference, not the one you cited in reference #6 by McCarthy et al, 2015). I did not have the chance to look at and critically appraise the research methods that were used to derive this statistic as the report only reports findings, not methods – hence, I find the statistic questionable. I encourage the authors to look at that original report. Finally, the authors should look at the more up-to-date report, Global Burden of Disease Study 2017 at [https://doi.org/10.1016/S2213-2600\(20\)30105-3](https://doi.org/10.1016/S2213-2600(20)30105-3), which suggests that the prevalence is higher in high-income countries, i.e.: “The global prevalence in 2017 was around 7.1% (95% UI 6.6–7.7). Chronic respiratory diseases were most prevalent across the GBD high-income super-region, at 10.6% (9.9–11.3), up from 9.7% (9.1–10.3) in 1990 (table 1). By contrast, the lowest prevalence was observed in sub-Saharan Africa (5.1% [4.5–5.8]) and south Asia (5.5% [5.1–6.0]). Latin America and the Caribbean had the largest decline (–0.80 percentage points) in chronic respiratory disease prevalence over the study period, from 8.1% in 1990 to 7.3% in 2017. Sub-Saharan Africa and the central Europe, eastern Europe, and central Asia super-region also saw declines in the prevalence of chronic respiratory diseases.” My assumption is that the higher the prevalence, the higher the burden. Of course, COPD burden includes its related “deaths” but we just need to be sure of the quality of the 2010 research methods that established, as a fact, that “more than 90% of COPD-related deaths occur in LMICs”. I mean, this is 2022, a decade has gone.

I think that the reason that this 90% figure is still being used, is its ability to concisely describe the problem, unfortunately the nuanced and more detailed statistics from the Global Burden of Disease are not easy to summarise. We have used the GDB reference and changed the statement:

Although most of these deaths occur in low and middle income countries (LMICs), most research on COPD management has been undertaken in high-income countries.

Authors should provide references for the following statements:

“The availability of pharmacotherapy is limited due to resource constraints, making pharmacological management difficult.”

“In addition to the direct health and healthcare burden, COPD negatively impacts the health-related quality of life (HRQoL) of people living with the condition”

These statements have now been referenced

Authors should cite peer-reviewed journal publication(s), not an institutional website, for the statement “There is growing evidence that pulmonary rehabilitation (PR) is an effective and cost-effective therapeutic intervention to improve COPD symptoms, patients’ quality of life and to reduce risk of premature deaths.”

The correct references have now been added.

The statement that “there are currently no PR services offered to patients in Georgia” may not be entirely true because I saw webpages of some Georgian hospitals that advertise PR services (e.g., <https://clinchmh.org/service/pulmonary-rehabilitation/> and <https://breathingbetter.org/our-services/pulmonary-rehabilitation/>). Should the authors rephrase their statement to indicate that there are few pockets of PR in Georgia (few but not completely absent), or that no PR studies exist on Georgia?

We can confirm that at the time of undertaking the research there were no PR services in Georgia; these URLs link to services with the US state of Georgia.

Overall, the introduction is well written and concise. It is interesting that the authors also bring the subject of a “culturally tailored” PR and I wonder what they mean by that. What is “cultural tailoring”? They could add a few lines to the introduction on this, i.e., what’s wrong with current PR packages that necessitated the researcher to adapt/tailor traditional PR to the Georgian setting? Such information would serve as a rationale for their decision to tailor/adapt it to the Georgian setting. Authors may check the publication at <https://doi.org/10.12688/wellcomeopenres.17702.1> on how authors elsewhere justified the need for their culturally adapted PR (introduction’s third paragraph).

We are grateful to the referee for the reference to this interesting preprint manuscript that has been put online since our article was submitted. We have drawn on the reviewer’s suggestions to describe reasons for adaptation and highlighted that the terms tailored and adaptation are inter-changeable in our paper.

However, evidence of the effectiveness of PR for COPD in patients from LMICs is extremely limited, barriers to provision include infrastructure for diagnosis, essential drug availability, lack of skilled health professionals, and overall healthcare priorities which limit management options¹² and few studies have adapted PR to the local context in LMICs.¹²⁻¹⁴

Methods:

Under “Study design and participants”, authors should add the word “dyspnoea” after MRC to indicate that their inclusion criterion “MRC score ≥ 2 ” was for dyspnoea and the MRC Dyspnoea Scale was used, without which the reader could confuse it with the score for muscular power assessment which also uses a type of MRC scale.

We have added dyspnoea to all references of the MRC score.

Under “Recruitment and randomisation”, write “... patients with COPD ...” not “... COPD patients ...”

Changed

Reference #17 (APEASE criteria) is a book chapter and should be cited as such (i.e., Michie S, Atkins L, West R. The APEASE criteria for designing and evaluating interventions. In: The Behaviour Change Wheel: A Guide to Designing Interventions. London: Silverback Publishing; 2014)

This has been changed.

Under “Intervention”, the authors write “The adapted PR programme drew on international guidance” and provide references 16 and 20 to support this, but reference 16 is not international as it all its authors are from the UK and the publication is specifically a British guideline.

We would argue that the ERS/ATS guideline is international and it is referred to within the same sentence.

Under “intervention”, authors should describe the exercises prescribed in the PR programme (e.g., type: strength/endurance/flexibility, lower limb/upper limb, warmup/cool down, interval/continuous training, exercise names/examples, etc)

We have added this to the supplementary file 1.

Under “intervention” the authors write “The programme took place twice weekly” and I am just wondering if they participants were advised to perform a third unsupervised home-based session as has been done in other studies.

Yes, participants were advised to do a third unsupervised exercise session (detail added to supplementary file 1).

Under “intervention”, what education topics were offered to the control group, and why were they given PR educational booklet?

The educational topics mirrored those of the intervention group and they were given the educational booklet as it covered information relevant to their self-management.

Under “Outcome measures”, authors write that “Data were collected from patients by the research team members at the end of the PR programme (eight weeks) and at six months”. Were data also not collected before the commencement of the programme (at week 0), and why was a 6-months follow-up assessment done while the program ended at week 8? Also in this section, mention the outcomes that were assessed, and the outcome measures used.

The baseline data collected is described in the section ‘Outcome measures’:

A baseline questionnaire captured sociodemographic and health-related characteristics of the participants.

Data were collected at 6-months to assess feasibility of follow-up and data collection. This is standard practice in feasibility trials.

Under “feasibility outcomes”, how did the authors define and assess “acceptability”?

<http://dx.doi.org/10.1186/s12913-017-2031-8>

We are aware of Sekhon's theoretical framework of acceptability. We assessed acceptability by objective measures of behaviour such as drop-out from the intervention and self-reported experience of the intervention.

The authors proceed to a "Secondary outcomes" section without having specified in the preceding sections what "primary outcomes" were. Or were the "feasibility outcomes" the primary outcomes? If yes, indicate them as such using explicit words "primary outcomes" as you did with "secondary outcomes", rather than making the reader guess.

Feasibility trials don't generally have a single primary outcome, it is normally 'the feasibility of intervention delivery and trial processes' – these are then normally assessed as criteria for progression to a full trial. This is the approach we have taken in this study and we have described these as the 'main feasibility outcomes'. We have defined what we consider the primary outcome of a future RCT would be (i.e. the SGRQ).

Under "Secondary outcomes", authors are not clear on the stage of the study at which the listed outcomes were assessed. For example, they are repetitive of the SGRQ in the second and third lines. In the second line, they write "at six months" and in the third line, they repeat the SGRQ and include both eight weeks and six months – confusing! Also, they write in the third line "SGRQ at programme end" – what was your programme? Is it at eight weeks or six months which you have already mentioned in the same third line before the colon? Also, the authors still do not mention whether they had assessed all the secondary outcomes mentioned in this section at baseline (at week 0). Again, why were the assessments repeated at six months when, it seems to me, the programme ended at eight weeks? Was it to assess the long-term impact of PR? Or you delivered a maintenance programme after eight weeks?

To clarify, we state that we will measure the *primary outcome of a future definitive RCT, which was the St George's Respiratory Questionnaire (SGRQ)²¹ at six months (total, symptoms, activity and impact)*. We then go on to list the other secondary outcomes. Clearly the primary outcome could only be measured at one time point so the SGRQ is repeated with 'at programme end' (i.e. 8 weeks) following it; all other outcomes are at 8-weeks and 6-months as specified.

Other secondary outcome measures at eight weeks and six months were: SGRQ at programme end, exercise capacity measured by the ISWT (metres),¹⁸ smoking status validated by cotinine,....

This is the most concise means of expression for this part of the text and was found to be acceptable by the other 3 reviewers, we therefore wonder if it would be better to avoid adding extra text.

Under "process evaluation", the authors write "The average interview duration was 35 minutes for the participants and 33 minutes for the rehabilitation specialists." Were the rehabilitation specialists not treated as study participants (i.e., consent etc)?

Indeed they were, we have added 'patients' in front of participants.

Under "Data analysis section", authors should not simply write that data were analysed using STATA/IC 15.1.3 – also tell us about the specific statistical tests you conducted in your data in STATA.

There wasn't any statistical testing in line with good practice for a feasibility trial.

I think the qualitative data analysis procedures done should be more comprehensively described.

We have kept this brief to keep within the word count. The qualitative findings presented are quite limited, and are described and cited in keeping with other papers of this nature.

Results

It is surprising that the authors said they aimed to recruit 60 participants and the flow diagram (figure 1) directly leads to this sample size. Please clarify in the methods section if 60 was planned before recruitment, or it was the number of eligible participants you were left with during pre-PR assessments, or it was by coincidence that what you planned before recruitment (60 patients) was the same sample size (60 patients during pr-PR eligibility assessments).

It wasn't a coincidence, we planned to recruit 60 participants, so stopped recruitment at this point. Our prospectively registered ISRCTN registration confirms the sample size to be 60. The methods state:

We aimed to recruit 60 participants. The sample size was chosen to enable estimation of feasibility outcomes with reasonable precision.¹² A follow-up rate of 80% could be estimated with a precision of 68% to 89% (binomial exact 95% confidence interval).

Authors wrote in the methods section that p values and 95% CIs were not reported but I see some 95% CIs being reported in the results section, e.g., "... with mean changes (SD) of -24.9 (25.4) [95%CI -40.3, -9.6] and 120.8m (89.5) respectively."

This statement refers to the secondary outcomes. A previous sentence states:

Although the trial was not powered to detect a difference between intervention and wait-list control, we calculated the mean change in SGRQ at PR end and 6 months for those allocated to each group; 95% confidence intervals were provided for estimates obtained.

As already commented elsewhere above, the authors did not tell us what pre-PR assessments were done at baseline (week zero). Those baseline/week zero measurements would have been included in Table 2, but they are not included in this table; hence the "mean changes" displayed in table 2 do not make sense since the reader/reviewer is not provided with baseline measurements with which to compare with measurements at 8 weeks and 6 months. Please display baseline measurements in table 2. Check the other tables too.

The baseline measurements are included in table 1. The change values are given in table 2 alongside the follow-up value. We are happy to add in the baseline values, but it will make the table very large and busy.

As an afterthought, were biometrics such as weight and BMI of participants measured? Would have been included in table 1 if important, especially also because I see one rehab specialists in the qualitative section reporting patients' weight gains in kilograms. Would this be an important outcome?

BMI and weight are not commonly reported for pulmonary rehabilitation. These were measured during the spirometry, but were not captured within the study database and they were not specified as secondary outcomes.

Outcome changes in the results section should have been reported against their minimal clinically important differences (MCIDs) available in literature (for ISWT distance, CAT, etc) since statistical significance tests couldn't be carried out.

The MCIDs have now been included within the footnotes to the tables (see earlier response) and then highlighted in the discussion.

I think the authors definition of "intervention fidelity" is too narrow. Is it really just about "increasing exercise prescriptions"? (Siedlecki, Sandra L. PhD, RN Research Intervention Fidelity, Clinical Nurse Specialist: 1/2 2018 - Volume 32 - Issue 1 - p 12-14 doi:

10.1097/NUR.0000000000000342. https://journals.lww.com/cns-journal/Citation/2018/01000/Research_Intervention_Fidelity_Tips_to_Improve.4.aspx#:~:text=Intervention%20fidelity%20means%20that%20the,internal%20validity%20of%20a%20study.)

We acknowledge that we have used quite a narrow definition of intervention fidelity.

I think that, due to journal word count limits, the authors haven't comprehensively reported data, especially the qualitative one (looking at the various questions contained in the interview topic guides vs the small number of participants' quotes in the results section – but I am more familiar with thematic analysis than content analysis, so the authors would know better). As a solution, other authors of a similar elsewhere split their data into two manuscripts (one for qualitative <https://doi.org/10.2147/COPD.S165623> and another for quantitative <https://doi.org/10.2147/copd.s146659>) and published these separately. But this would perhaps be considered as unethical by other editors (e.g., read about salami publication at <https://dx.doi.org/10.11613%2FBM.2013.030>) and I leave this to the BMJ Open editors to decide.

The qualitative data was relatively limited, and with a sample size of just nine participants not rich enough for a full paper. Additionally, the qualitative interviews were focussed on the process evaluation, so we feel they fit better to illustrate domains of the feasibility study not addressed quantitatively.

Discussion

As commented above, results should have been discussed in relation to published MCIDs where available (I know ISWT has an MCID, and I am sure the other outcome measures also have). The statement “Participants who attended demonstrated improvements in clinical outcomes” would confuse the reader as they wouldn’t know whether you are talking of statistically significant improvement or clinically important improvements.

We have amended the sentence to address this:

Participants who attended demonstrated clinically significant improvements in outcomes, exceeding the minimally important clinical differences for the SGRQ, PHQ-9, GAD-7 and ISWT.

VERSION 2 – REVIEW

REVIEWER	Faniel Bickton Malawi-Liverpool-Wellcome Trust Clinical Research Programme, Lung Health Research Group
REVIEW RETURNED	07-Jun-2022
GENERAL COMMENTS	I thank the authors for responding to most of my issues previously raised.